# Impact of Assimilating GK-2A All-Sky Radiance with a New Observation Error for Summer Precipitation Forecasting

**Miranti Indri Hastuti [1,2]** and **Ki-Hong Min [1,*]**

1   Department of Atmospheric Sciences, Kyungpook National University, Daegu 41566, Republic of Korea; miranti.hastuti@bmkg.go.id
2   Kualanamu Meteorological Station, The Agency for Meteorology, Climatology, and Geophysics of the Republic of Indonesia (BMKG), Jl. Tengku Heran, Beringin, Deli Serdang 20552, Indonesia
*   Correspondence: kmin@knu.ac.kr

**Abstract:** In the assimilation of all-sky radiance (ASR), the non-Gaussian behaviour of observation-minus-background (OMB) departures has been the major issue. Treating observation error properly should give the distribution OMB departures closer to Gaussian on which data assimilation systems are based. This study introduces a look-up-table (LUT) observation error inflation (LOEI) for assimilating ASR from three water vapor channels of GEO-KOMPSAT-2A (GK-2A) geostationary satellite based on a three-dimensional variational data assimilation (3DVAR) framework. The impacts are assessed based on summer precipitation cases over South Korea. To confirm all kinds of radiance observations, the ASRs are assimilated without any quality control procedures. The LOEI adopt a pre-estimated radiance error statistics by using the higher order fitting function of cloud amount ($C_A$) and standard deviation (STD) of OMB departures. This LOEI was produced during the summer period from August 1 to 30, 2020, representing the characteristics of the atmosphere condition during the experimental period. The promising impact of LOEI is demonstrated in comparison with the inflated observation error using a simple linier function proposed by Geer and Bauer (GBOEI). Study results revealed the LOEI normalized OMB departures into much more Gaussian form than the GBOEI. Hence, the assimilation of ASR using LOEI (ExpLOEI) produced BT analysis closer to the observation in four cloud phases in contrast with ASR assimilation using GBOEI (ExpGBOEI), which obviously found in the ice phase. The better BT analysis eventually simulated more realistic moisture and temperature variables in the background field. Consequently, the ExpLOEI exhibited more accuracy in precipitation location and intensity compared to the experiment with ExpGBOEI.

**Keywords:** GK-2A; ASR; observation error; data assimilation; rainfall forecast

## 1. Introduction

Progress on the use of satellite observations at numerical weather prediction (NWP) centers has come largely rapid in recent years [1]. Many different types of satellite observations data are utilized into NWP data assimilation systems [2,3]. Among these types of observations data, the brightness temperature (BT; determined as "radiance") from infrared channels has significantly improves NWP performance [4–6]. However, radiances are quite limited usage which only clear-sky radiances (CSR) has been extensively used in data assimilation [7–9]. Indeed, assimilating CSR improves the analysis of mass and thermal state (pressure, temperature, and wind), increasing the forecast accuracy at NWP centers [10]. But the prominent benefits of radiance data can be optimized through the assimilation of the all-sky radiance (ASR) [11]. In NWP systems, ASR is also sensitive to moisture, clouds, and precipitation fields, which is an essential information for predicting the summer precipitation [12].

To assimilate ASR, such issues may arise to severely degrade the quality of analysis [13–15]. Previous studies reported the non-Gaussianity of observation-minus-background (OMB)

departures majorly hampers the assimilation of ASR, which was frequently caused by the deficiencies of the radiative transfer model (RTM) in simulating the ASR background (especially cloudy radiance) [16]. The RTM encounters much difficulties associated with moist physics processes and vertical structures of cloud parameters [17]. These problems also lead to discontinuity between clear- and cloudy-radiances in the initial conditions [18,19]. Among the cloud types, the simulation of thick ice clouds is immensely more difficult than other types of clouds due to the insufficiency of cloud condensation in RTM, causing the underestimation of thick ice clouds simulation [20]. Due to those systematic issues of simulating ASR by RTM, the characteristics of OMB departures in ASR assimilation can be understood, thus observation error for ASR can be modelled [21]. A proper estimation of ASR observation error can normalize the distribution OMB departures to be close to Gaussian distribution on which most data assimilation systems are assumed [22].

Geer and Bauer (2011) (GB11) examined the systematic difference of OMB departures in microwave imager ASR, demonstrating that ASR have a heteroscedasticity characteristic. This means that errors are larger in rainy and cloudy situations but lower in clear situations [23]. Hence, observation error must be not universal but vary spatially. Inflating the observation errors substantially determines the weight of ASR observations, which solves the fat-tail distribution in the initial OMB departures. In addition, GB11 showed the variations in the standard deviation (STD) of OMB departures can be described by a cloud amount parameter ($C_A$). Using $C_A$, GB11 formulated a stepwise linier function with parameter thresholds to calculate observation error inflation for ASR assimilation (GBOEI).

Using the GBOEI method, a few studies on improving ASR data assimilation have been conducted. Okamoto (2014) introduced a new $C_A$ equation for infrared atmospheric sounding interferometer (IASI) satellite, which also can simulate the distribution of STD of OMB departures [24]. Further, Okamoto et al. (2017) tested these new-defined $C_A$, developed quality control procedures (QCs), and predicted observation error using GBOEI method for ASR data assimilation from geostationary Himawari-8 satellite [25]. Results showed that normalized OMB distribution was Gaussian for water vapor channels in the most cases in which QCs and cloud-dependent observation error were used, leading to better analysis variable and forecasts. Harnish et al. (2016) estimated ASR observation errors using the new robust $C_A$, which treats cloud-free and cloud-affected radiances in the uniform way [26]. Xu et al. (2016) applied the channel-dependent observation errors for satellite Himawari-8 ASR data assimilation based on three-dimensional variational framework (3DVAR), showing that simulated ASR fits to observation and improved cloud heights [27].

Those above studies showed that GBOEI has brought great benefits on the ASR assimilation. However, the uncertainties of giving proper thresholds in GBOEI has remained a big task. Moreover, according to Waller et al. (2017), ASR observation error should be estimated from statistical technique, not calculated [28]. It has been proven that the observation error method introduced by Desroziers et al. (2005), which are based on statistical estimation, is becoming popular due to its applicability and effectivity [29]. The research on predicting ASR error statistics from GEO-KOMPSAT-2A (GK-2A) is also largely unexplored. This satellite is South Korea's second geostationary meteorological satellite stationed at 128.2° East and was launched on 4 December 2018. The GK-2A satellite observations has different frequency channels which frequently contains correlated errors and different characteristics. Hence, applying the channel-dependent ASR observation error statistics is crucial [30].

This study examines the ability of a look-up-table (LUT) observation error inflation (LOEI) estimated from statistical average of OMB residuals for each water vapor channels to improve the assimilation of ASR from GK-2A satellite. The impacts were investigated based on the two storm cases associated with summer Changma front using the high-resolution weather research forecasting (WRF) model and three-dimensional variational data assimilation (3DVAR) framework. In comparison, the ASR assimilation using LOEI and the existing linier-function of GBOEI were displayed to analyze the differences and impact on the summertime heavy precipitation forecasts.

## 2. Data and Experimental Methods

### 2.1. Observation Data

The GK-2A is operated by the National Meteorological Satellite Center (NMSC) of the Korea Meteorological Administration (KMA). For conducting observations, the GK-2A has an instrument on-board called Advanced Meteorological Imager (AMI) with 16 channels which is similar to Advanced Himawari Imager (AHI) aboard the Japanese Himawari-8 and Advanced Baseline Imager (ABI) aboard the United States Geostationary Operational Environmental Satellite Citation 16 (GOES-16). The GK-2A AMI observation captures the Earth in 16 different wavelengths of light, in which 4 wavelengths are visible light and 12 wavelengths are various combinations of near-infrared and medium/longwave infrared. The GK-2A AMI scans the earth full disk every 10 min and the Korean Peninsula every 2 min with 2 km resolution for infrared (IR) channels. The full specifications of GK-2A AMI is listed on the Table 1. The water vapor channels are sensitive to the humidity in the middle and upper troposphere [31]. Thus, this study utilized the water vapor bands (6.3, 6.9, and 7.3 μm or channel 8, 9, 10) for ASR data assimilation. NMSC has developed the algorithms to retrieve the a few kinds of specific product which refers to level-2 products. There are 52 products are classified into 23 primary products and 29 side products. One of the 23 primary products is level-2 cloud phase product which used on this study to classified individual sky types.

**Table 1.** The specifications of GK-2A AMI.

| Channel Number | Channel Name | Wavelength (μm) | Resolution (km) | Observation Characteristics |
|---|---|---|---|---|
| 1 | VI004 | 0.4708 | 1 | |
| 2 | VI005 | 0.5068 | 1 | |
| 3 | VI006 | 0.6394 | 0.5 | |
| 4 | VI008 | 0.8630 | 1 | Land and sea masks and vegetarian |
| 5 | NR013 | 1.3740 | 2 | Cloud physical parameter |
| 6 | NR016 | 1.6092 | 2 | |
| 7 | SW038 | 3.8316 | 2 | Low-level clouds, fog, wildfires |
| 8 | WV063 | 6.2104 | 2 | Vertical humidity profile (middle-to-upper tropospheric level) |
| 9 | WV069 | 6.9413 | 2 | |
| 10 | WV073 | 7.3266 | 2 | |
| 11 | IR087 | 8.5881 | 2 | Thin ice cloud monitoring |
| 12 | IR096 | 9.6210 | 2 | Ozone absorption |
| 13 | IR105 | 10.3539 | 2 | Ice crystals/water, lower water vapor, volcanic ash, sea surface temperature |
| 14 | IR112 | 11.2288 | 2 | |
| 15 | IR123 | 12.3664 | 2 | |
| 16 | IR133 | 13.2908 | 2 | $CO_2$ absorption, cloud top height |

The automatic weather stations (AWS) precipitation observations operated by KMA were used as the verification data. There are 969 AWS observations (Figure 1) which were interpolated into the same domain size and resolution of model grid used on this study. Furthermore, the Osan radiosonde observation was also utilized to verify the water vapor mixing ratio and temperature parameter on the forecast models (Figure 1).

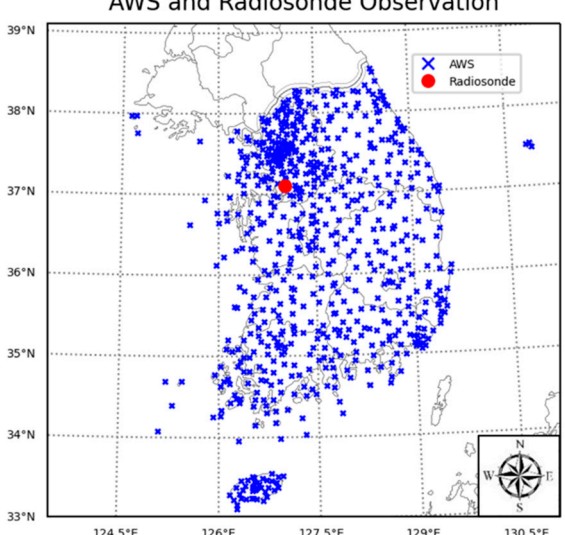

**Figure 1.** Locations of the AWS (blue) and Osan radiosondes (red) observation sites operated by KMA.

### 2.2. ASR Assimilation Method

#### 2.2.1. The LOEI Method

Among the challenges for assimilating ASR effectively, applying observation errors on each radiance observations has an important impact on the quality of the analysis since the determination of the weights for the observation is directly influenced by observation error in the analysis procedure. The observation errors should be situation-dependent, considering the representativeness errors that are smaller in clear-sky regions and larger in the presence of clouds and precipitation, and should be applied as channel-dependent since each channel has different characteristics and sensitivity detection [23]. Okamoto et al. (2014) developed a parameter for expressing the magnitude of $C_A$ based on observed and simulated ASR that can describe the variations of STD of OMB departures. $C_A$ can be written as Equation (1) [24]:

$$C_A = \frac{(|B - B_{clr}| + |O - B_{clr}|)}{2} \tag{1}$$

where $C_A$, O, B are cloud amount parameter, observation, and simulated ASR, respectively. $B_{clr}$ is simulated CSR when the cloud scattering is switched off in CRTM.

The ASR observation error can be defined as the STD of OMB departures. The $C_A$ can be utilized to determine the STD of OMB departures; hence, $C_A$ able to predict the observation error. GB11 developed the GBOEI with their symmetric cloud parameter to predict the STD of OMB departures and estimate the observation errors, as expressed in Equation (2) [23]. These estimations applied an increased observation errors for a larger $C_A$ and a constant observation error for larger $C_A$ after reaching a maximum of STD of OMB departures.

$$\begin{array}{ll} GBOEI = STD_{min} & for\ C_A \leq C_{A\_min} \\ GBOEI = STD_{min} + \frac{STD_{max} - STD_{min}}{C_{A\_max} - C_{A\_min}}(C_A - C_{A\_min}) & for\ C_{A\_min} < C_A \leq C_{A\_max} \\ GBOEI = STD_{max} & for\ C_A \geq C_{A\_max} \end{array} \tag{2}$$

where $STD_{min}$, $STD_{max}$, $C_{A\_min}$, and $C_{A\_max}$ are the thresholds parameters. The $STD_{min}$ and $STD_{max}$ are the minimum and maximum of STD, respectively. The $C_{A\_min}$ and $C_{A\_max}$ are the minimum and maximum of $C_A$, respectively.

For LOEI, the $C_A$ and STD of OMB statistics was calculated from ASR samples without any QCs in August 2020 every 6 h. The non-QCs was applied to validate all kinds of all-sky

observations. The observation error dependance on $C_A$ as in GBOEI was also applied. This LOEI adopts a LUT for $C_A$ bins of 1 K, and the STD of OMB for each bin was computed separately. Then, $C_A$ bins and the STD of OMB were fitted using higher-order polynomial regression, which is written as Equation (3):

$$y = ax + bx^2 + cx^3 + dx^4 + ex^5 + f \qquad (3)$$

The $y$, as the polynomial fit, represents the observation error. The $a$, $b$, $c$, $d$, $e$, and $f$ are the coefficients. The $x$ represents the STD of OMB. Before reaching a maximum STD of OMB value, observation error follows these polynomial fit values and the rest remain constant with the highest value of polynomial fit. The flowchart of LOEI model is displayed in Figure 2.

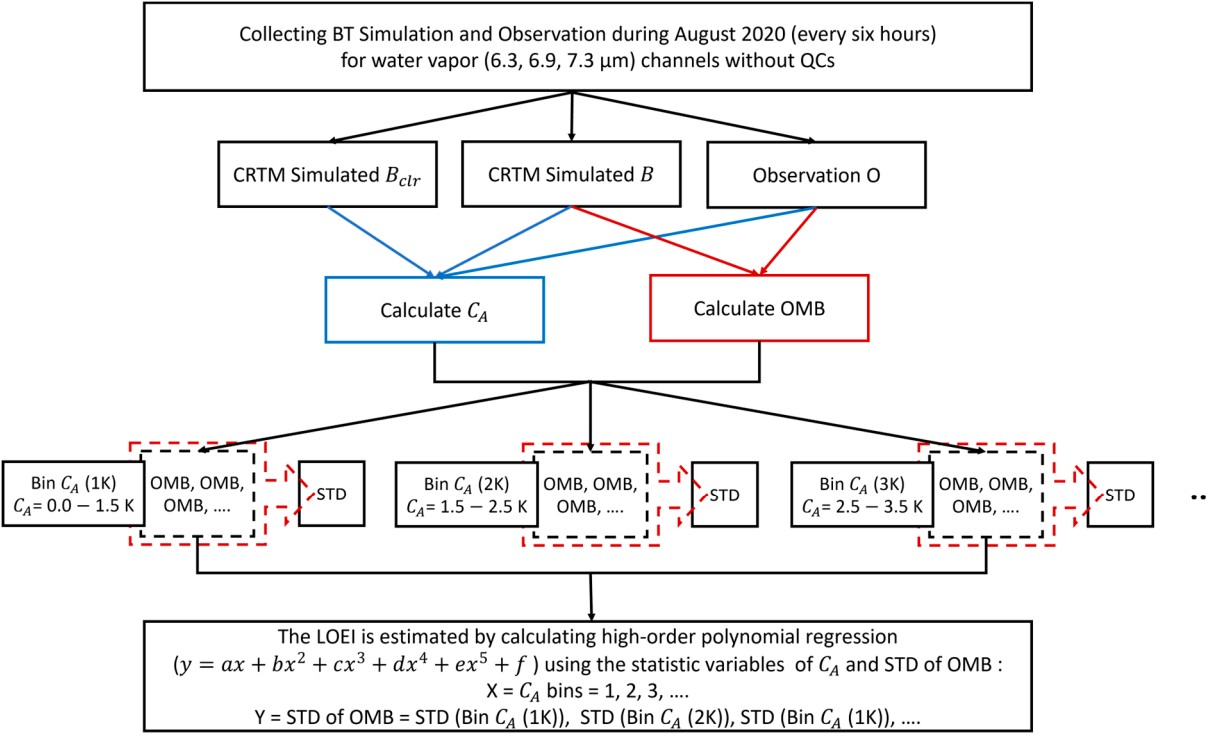

**Figure 2.** The flowchart of LOEI method.

### 2.2.2. Observation Operator

Currently, the WRF data assimilation (WRFDA) system interfaces with the two most popular fast RTMs: Radiative Transfer for Television Infrared Observation Satellite Operational Vertical Sounder (RTTOV), developed and maintained by the European Organization for the Exploitation of Meteorological Satellites (EUMETSAT), and the Community Radiative Transfer Model (CRTM) [32], developed by the United States Joint Center for Satellite Data Assimilation (JCSDA) [33]. In this study, CRTM is used as the observation operator for calculating simulated radiance data from the model state vector. CRTM converts the radiative transfer problem into various components: surface emissivity/reflectivity model, aerosol and cloud absorption scattering model, and gaseous absorption model. The inputs of the CRTM are pressure, temperature, water vapor, and water content of six hydrometeor types (rain, snow, ice, graupel, and hail) from the background. The general RTM

equation assumes plane parallel, non-polarized atmosphere, and vertically-stratified, the monochromatic radiance can be written as Equation (4) [33]:

$$\frac{dI(\tau;u,\phi)}{d\tau} = -I(\tau;u,\phi) + (1-\omega)B(T) + \frac{\omega}{4\pi}\int P(\tau;u,\phi;u',\phi')I(\tau;u',\phi')du'd\phi'$$
$$+\frac{\omega}{4\pi}P(\tau;u,\phi;-u_\otimes,\phi_\otimes)F_\otimes e^{\frac{-\tau}{u_\otimes}} \tag{4}$$

where $I,\tau,B,P,\omega$ represent intensity, optical depth, Planck function, phase function, and single-scattering albedo, respectively. In the direction of incoming $(u',\phi')$ and outgoing $(\mu,\phi)$ light beams, where $\mu' = \cos(\theta')$ and $\mu = \cos(\theta)$, $\theta'$ and $\theta$ are the zenith angle and $\varphi'$ and $\varphi$ the azimuth angle. $F_\otimes$ represents the solar irradiance incident in the direction $(-u_\otimes,\phi_\otimes)$, where the minus sign indicates downward propagation. Term A is negative due to attenuation of radiation by extinction. In the right hand side of Equation (4), the first two terms are positive due to an increase in the radiation provided by the source functions. The last term can be ignored in the microwave and infrared spectral range. The CTRM has a solver that solves the RTMs equation for given atmospheric optical depth profile, surface emissivity and reflectivity, cloud optical parameters, and source functions. CSR simulation does not involve scattering, in which the the third term can be ignored, yielding the Schwarzschild's equation. For all-sky cases, the radiative transfer is solved with the advanced doubling adding method [34].

### 2.2.3. Variational Bias Correction

To successfully assimilate radiance data, systematic bias must be corrected before assimilation [35–37]. The biases arise as a result of systematic errors in any one of following sources: instrument calibration, RTM, and background atmospheric state provided by the NWP model used for monitoring. The bias correction model modifies observation operator with a linear combination of set predictors (Equation (4)):

$$\widetilde{H}(x,\beta) = H(x) + \beta_0 + \sum_{i=1}^{N_p} \beta_i P_i \tag{5}$$

where $H$ represents the original operation operator, $x$ is the model state vector, $\beta_0$ is a constant component of total bias, and $P_i$ and $\beta_i$ are the $i$th of $N_p$ predictors and corresponding bias correction coefficients, respectively. The bias correction can be estimated off-line for each channel [38] or updated adaptively within variational minimization process by including them in the state vector [39]. The latter method is called variational bias correction (VarBC). In WRFDA, VarBC includes seven predictors: the scan position, square and cube scan position, 50–200 and 300–1000 hPa layer thicknesses, surface skin temperature, and total column water vapor. The index of the pixel in the field view is the scan position for polar-orbiting satellites, and the satellite zenith angle is the scan position for geostationary satellites. On adding the radiance VarBC, $J$ is to be minimized with the inclusion of the background states and bias parameter (Equation (5)):

$$J(x,\beta) = \frac{1}{2}(x-x_b)^T B^{-1}(x-x_b) + \frac{1}{2}(\beta-\beta_b)^T \beta_b(\beta-\beta_b)$$
$$+\frac{1}{2}\left[y-\widetilde{H}(x,\beta)\right]^T R^{-1}\left[y-\widetilde{H}(x,\beta)\right] \tag{6}$$

$\beta$ and $\beta_b$ represent the correction coefficient vector of background bias and the associated error covariance, respectively. The bias correction coefficients of the predictors are updated along with the variational analysis during the minimization procedure using coefficients from previous cycles' analysis as the background for every analysis.

### 2.3. WRF 3DVAR Assimilation System

This study conducted assimilation based on the 3DVAR system from WRFDA, version 4.2 [40]. The 3DVAR system merges background information and observations for the initial state and utilizes a linearized prediction model to establish dynamic and realistic observations with an accurate analysis and forecast field [41]. The 3DVAR system produces a better initial condition through an incremental approach, minimizing the cost function $J(x)$ for finding the analysis state $x$ [42], as defined by Equation (1).

$$J(x) = J_b(x) + J_o(x) = \frac{1}{2}(x - x_b)^T B^{-1}(x - x_b) + \frac{1}{2}(y_0 - H(x))^T R^{-1}(y - H(x)) \quad (7)$$

The $J_o(x)$ and $J_b(x)$ represent the cost functions obtained from observation and background, respectively. $x$, $x_b$, $y_0$, $H$, $B$, and $R$ represent the analysis field, first guess, observation, nonlinear observation operator, background error (BE) covariance matrix, and observation error covariance matrix, respectively. $B$ matrix is extremely huge ($10^7 \times 10^7$) and calculating its inverse $B^{-1}$ is difficult. Hence, the matrix decomposition ($B = UU^T$) is applied for simplification. U is decomposed of B. The transformation of control variable $v$ with the assumption ($Uv = x - x_b$) is commonly applied. Therefore, Equation (1) can be transformed into Equation (2):

$$J(x) = J_b(x) + J_o(x) = \frac{1}{2}v^T v + \frac{1}{2}(d - H'Uv)^T R^{-1}(d - H'Uv) \quad (8)$$

The vector $d = y_o - H(x^b)$ is the innovation vector that measures the departure of observation $y_o$ from the background $x^b$, and $H'$ is the linearization of the nonlinear observation operator $H$. After these transformations, the $B$ matrix is implicitly given in the control variable operator and is not required to be presented directly.

The National Meteorological Center (NMC) method was used to produce the covariance matrix for the regional BE statistics [43] by taking 12 h and 24 h forecasts during the summer period from 1 August to 30 September 2020, representing the characteristics of the atmosphere condition during the experimental period. The BE statistics were calculated on the selected domain with the following five control variables: eastward velocity (U), the northward velocity (V), temperature (T), surface pressure (Ps), and pseudo relative humidity (RHs).

### 2.4. Model Configuration and Experimental Design

WRF features with a dynamic solver of the advanced research WRF [44] was utilized to construct the NWP model. The WRF model is completely non-hydrostatic and compressible and possesses a mass coordinate system. The WRF model serves a wide range of spatial scales, which is suitable for a broad span of meteorological applications. The WRF model contained three nesting domains (D01, D02, D03) shown in Figure 3. The resolutions of the three nesting domains were 9 km, 3 km, and 1 km, respectively. In the vertical level, it consisted of 60 η model layers, and the top of model was 50 hPa. The NCEP/FNL (National Centers for Environmental Prediction/Final Analysis) were used as the base of initial and boundary conditions, with a resolution of $1° \times 1°$ produced by the National Centers for Environmental Prediction/National Centers for Atmospheric Research (NCEP/NCAR). The forecast model simulations applied Kain-Fritsch Scheme for domains 1 and 2 as the cumulus parameterization [45]. For cloud microphysics, the WRF Double Moment 6 class (WDM6) scheme was employed [46]. The Rapid Radiative Transfer Model (RRTM) longwave–Dudhia shortwave schemes [47], Unified Noah land surface model [48], and the YSU scheme [49], were applied for the atmospheric radiation process, land surface, and planetary boundary layer, respectively. All model physics and vertical levels were the same in all domains. The model configurations are summarized in Table 2.

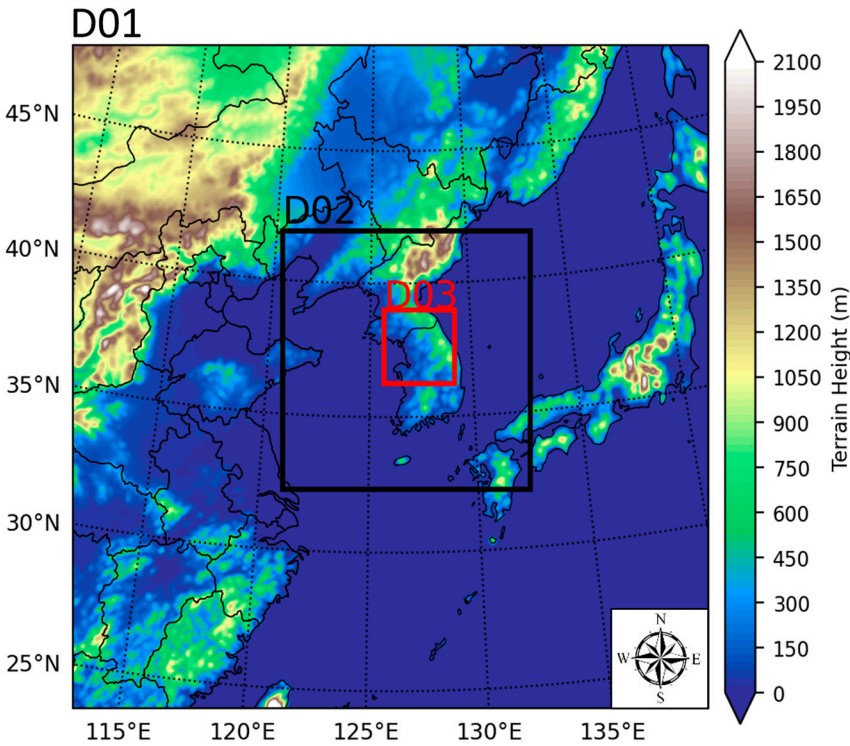

**Figure 3.** Domain configuration and topography (shaded) of D01, D02, and D03.

**Table 2.** The Weather Research and Forecasting (WRF) model configurations.

|  | **D01** | **D02** | **D03** |
|---|---|---|---|
| WRF version | | v4.2 | |
| Resolution | 9 km | 3 km | 1 km |
| Horizontal Grids | 301 × 301 | 352 × 352 | 301 × 301 |
| Vertical Grids | 60 | 60 | 60 |
| Cumulus | | Multiscale Kain–Fritsch scheme | |
| Microphysics | | WRF Double Moment 6 class scheme | |
| Planetary Boundary Layer | | Yonsei University Scheme | |
| Surface Layer | | Revised MM5 Monin–Obukhov scheme | |
| Land Surface | | Unified Noah land surface model | |
| Radiation | | Rapid radiative transfer model scheme Long-wave/Dudhia Scheme Short-wave | |
| Initial and Boundary Conditions | | NCEP FNL 0.1 Degree Global Tropospheric Analysis | |

Three experiments are conducted to understand and analyze the impact of LOEl for GK-2A satellite ASR assimilation in improving the summertime heavy storms in Korea (Table 3). The CTRL was the simulation without data assimilation. The ExpGBOEI and ExpLOEI experiments assimilates ASR using GBOEI and LOEI, respectively. Both experiments assimilate satellite ASR into D01, D02, and D03. The assimilation time was conducted during a 3 h period with continuous cycling in a 30 min window (Figure 4).

**Table 3.** Experimental design.

| Experiment | Assimilation | Observation Error |
|:---:|:---:|:---:|
| CTRL | N/A | N/A |
| ExpGBOEI | ASR | GBOEI |
| ExpLOEI | ASR | LOEI |

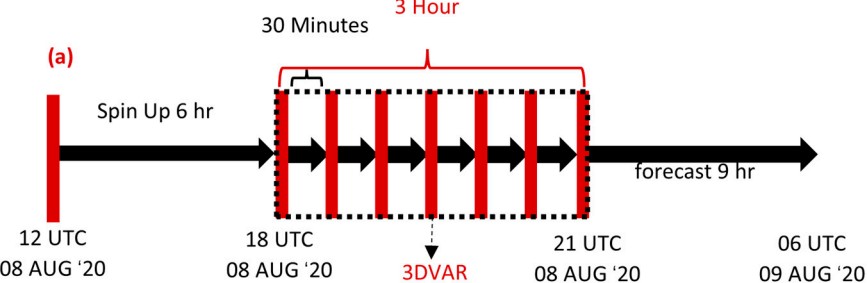

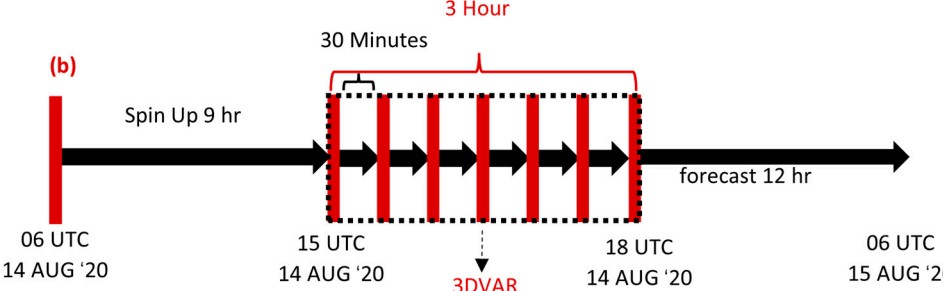

**Figure 4.** Schematic diagram of the experimental design illustrating the data assimilation cycle and forecast for (**a**) Case 1 and (**b**) Case 2.

### 2.5. Overview of the Cases

Two cases of heavy rainfall associated with convective systems along the Changma front during summertime in Korea were selected to investigate the performance of LOEI for ASR data assimilation. The two characteristics and forecast periods of selected cases are shown in Table 4.

**Table 4.** Selected cases for the numerical experiments and their storm characteristics.

| | Forecast Period | Total Cumulative Precipitation (mm) | Maximum Rain Rate (mm·h$^{-1}$) |
|:---:|:---:|:---:|:---:|
| Case 1 | 8 August 2020 at 2100 UTC—9 August 2020 at 0600 UTC | 107.32 | 53.0 |
| Case 2 | 14 August 2020 at 1800 UTC—15 August 2020 at 0600 UTC | 114.8 | 44.0 |

On 9 August 2020 (Case 1), the upper-level trough was observed at 500 mb over Mongolia in the northwest of the Korean Peninsula (Figure 5a), inducing Changma front over South Korea (Figure 5b). The Changma front propagated from southwest to northeast near Gyeonggi Province, and the low-level jet (purple zones) brought a huge amount of water vapor from the East China Sea into Yellow Sea (Figure 5b). Over the Yellow Sea, the moisture transport and convergence ($\pm 35 \times 10^{-9}$ s$^{-1}$) influence the formation of frontal systems (not shown), leading to heavy mesoscale rainfall over Gyeonggi-do and Gangwon-do Provinces (Figure 5c). The hourly rainfall recorded 53.0 mm in Osan, Gyeonggi-do

Province. The peak after 9 h precipitation reached 107.32 mm in Pogog-eup, Gyeonggi Province (Table 4).

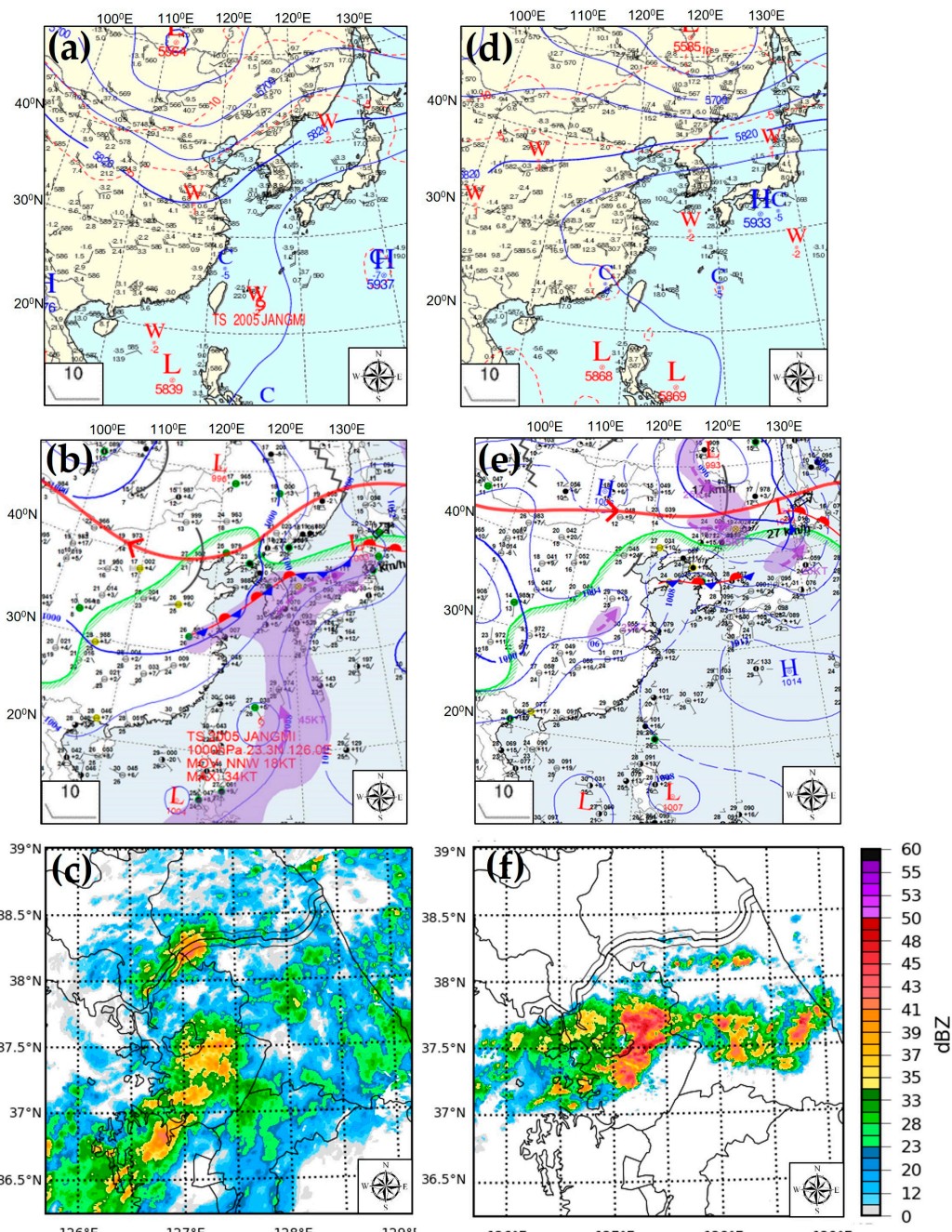

**Figure 5.** 500 hPa Weather Chart and synoptic analysis (**a**,**b**) at 0000 UTC on 9 August 2020 (Case 1) and (**d**,**e**) at 0000 UTC on 15 August 2020 (Case 2). Radar reflectivity (dBZ) of 1.5 km at (**c**) 2300 UTC on 8 August 2020 (Case 1) and (**f**) 0600 UTC on 15 August 2020.

On 15 August 2020 (Case 2), the weather map 500 mb showed high pressure over North Pacific that extended westward and low pressure over Northeast China that extended southward (Figure 5d), causing the Changma front to be propagated eastward over South Korea (Figure 5e), resulting in convective systems (Figure 5f). The convergence ($\pm 35 \times 10^{-9}$ s$^{-1}$) occurred over the Yellow sea, and the moisture transport brought large amount water vapor to Gyeonggi-do and Gangwon-do Provinces (not shown). The peak of hourly rainfall reached 44.0 mm in Gangnam, Gyeonggi-do Province. The peak after 12 h of precipitation reached 114.80 mm in Hoengseong, Gangwon Province (Table 4).

### 2.6. Evaluation Parameters

This study primarily aimed to enhance the precipitation prediction by assimilating ASR data assimilation with the LOEI. The verification score of the precipitation forecasts was utilized for evaluating the performance of data assimilation models. The quantitative error was verified using the hourly and cumulative rain of the forecast model against AWSs. Bias and root mean square error (RMSE) can be defined based on following equations [50]:

$$Bias = \frac{1}{N} \sum_{i=1}^{N} (P_i - O_i) \tag{9}$$

$$RMSE = \frac{1}{N} \sum_{i=1}^{N} (P_i - O_i)^2 \tag{10}$$

$N$, $P_i$, and $O_i$ represent the total number of observation data, rainfall prediction, rainfall observation, respectively. Bias determines whether the simulated forecast system tends to underpredict (BIAS < 1) or overpredict (BIAS > 1) events. RMSE indicates the average magnitude of the forecast errors with negatively-oriented scores and lower values being better.

The classification of the rainfall occurrences score was also performed to verify the quality of the forecast model. To verify this type of forecast, a contingency table (Table 5) is required to compute the frequency of 'yes' and 'no' forecasts and occurrences. Classified variables hits, false alarms, misses, and correct negatives represent whether the predicted and observed values in a certain observation period or position are greater than certain thresholds.

**Table 5.** Rain/no rain contingency table for categorical verification.

|  |  | Observation | | Total |
|---|---|---|---|---|
|  |  | **Yes** | **No** |  |
| Forecast | Yes | Hits | False alarms | Forecast Yes |
|  | No | Misses | Correct negative | Forecast No |
|  | Total | Observed Yes | Observed No | Total |

The accuracy, critical success index (CSI), and equitable threat score (ETS) were computed from the elements in the contingency table to describe particular aspects of forecast performance. The accuracy determines the proportion of correct predictions. The best accuracy score is one. The accuracy can be defined using the following equation (Equation (11)) [51].

$$Accuracy = \frac{Hits + Correct\ Negatives}{Total} \tag{11}$$

CSI measures the fraction of observed and/or forecast precipitation that was correctly predicted [52]. The fraction of precipitation forecast is sensitive to hits and penalizes both misses and false alarms. The value of one is the perfect score.

$$CSI = \frac{Hits}{Hits + Misses + False\ Alarms} \tag{12}$$

The ETS is similar to CSI, but ETS also calculates adjusted hits associated with random chance (Equation (14)) [52]. ETS is sensitive to hits since it penalizes both misses and false alarms in the same way, and it does not distinguish the source of forecast error. The highest accuracy of ETS is one.

$$Hits_{random} = \frac{(Hits + Misses)(Hits + False\ Alarms)}{Total} \tag{13}$$

$$ETS = \frac{Hits - Hits_{random}}{Hits + Misses + False\ Alarms - Hits_{random}} \tag{14}$$

To verify the pattern of accumulative precipitation, the pattern correlation between AWS data and the experimental results was calculated. The pattern correlation can be computed using the following equation [53]:

$$R_{patt\_cor} = \left| \frac{\sum_{i=1}^{N} (X_{obs,i} - \bar{X}_{obs})(X_{pred,i}\bar{X}_{pred})}{\sigma_{obs}\sigma_{pred}} \right| \tag{15}$$

$N$ represents the total number of observation data. $X_{obs}$ and $\bar{X}_{obs}$ represent the rainfall observation data and the mean of rainfall observation data, respectively. $X_{pred}$ and $\bar{X}_{pred}$ represent the rainfall prediction and the mean of rainfall prediction, respectively. $\sigma_{obs}$ and $\sigma_{pred}$ are the standard deviation of rainfall observation and prediction data, respectively. The score of pattern correlation ranges from zero to one. The value of one means both data are utterly linearly correlated. The value of zero means both data are not linearly correlated.

## 3. Results

### 3.1. ASR Observation Error

The LOEI were derived from higher-order fitting function between $C_A$ bins and the STD of OMB. Figure 6 shows the STD of OMB, GBOEI, and LOEI as a function of $C_A$. It can be inferred the characteristic of $C_A$ and STD of OMB relation. The smaller $C_A$ value means both observed and simulated BT are classified as CSR, associated with smaller error. While, the higher $C_A$ means there is a mismatching of sky radiance type between observed and simulated BT which produces a large OMB departure and requires higher error for the data assimilation. The STD of OMB departures peaks at 15.85 K in a $C_A$ bin of 16 K for channel 8, at 22.05 K in a $C_A$ bin of 20 K for channel 9, and at 26.96 K in a $C_A$ bin of 26 K for channel 10 (Figure 6). The GBOEI was able to model the dependence of $C_A$ and observation error. However, there was the value of GBOEI which smaller than the real value of STD of OMB departures before the peak value of STD of OMB. In addition, there was a uncertainty of determining the parameter thresholds for $STD_{min}$, $STD_{max}$, $C_{A\_min}$, and $C_{A\_max}$. Meanwhile, the LOEI fitted the real value of STD of OMB and avoid the uncertainty of thresholds as in linier stepwise function of GBOEI.

The data assimilation system assumes the Gaussian distribution between observation and background. The OMB departures can be normalized by using the observation error which leads to the Gaussian distribution; hence, a better analysis increments can be produced. The probability density function (PDFs) of normalized OMB by GBOEI (blue line) and LOEI (red line) was presented using OMB samples from 1–30 August 2020 every 6 h (Figure 7). Normalization with the GBOEI formed a too-pronounced peak of PDFs and larger deviations on the tails distribution for all water vapor channels compared to Gaussian distribution. Meanwhile, applying the LOEI for the OMB normalization reduced the strong PDFs peak and offset-tails distribution which led to a better fit to Gaussian forms in all water vapor channels.

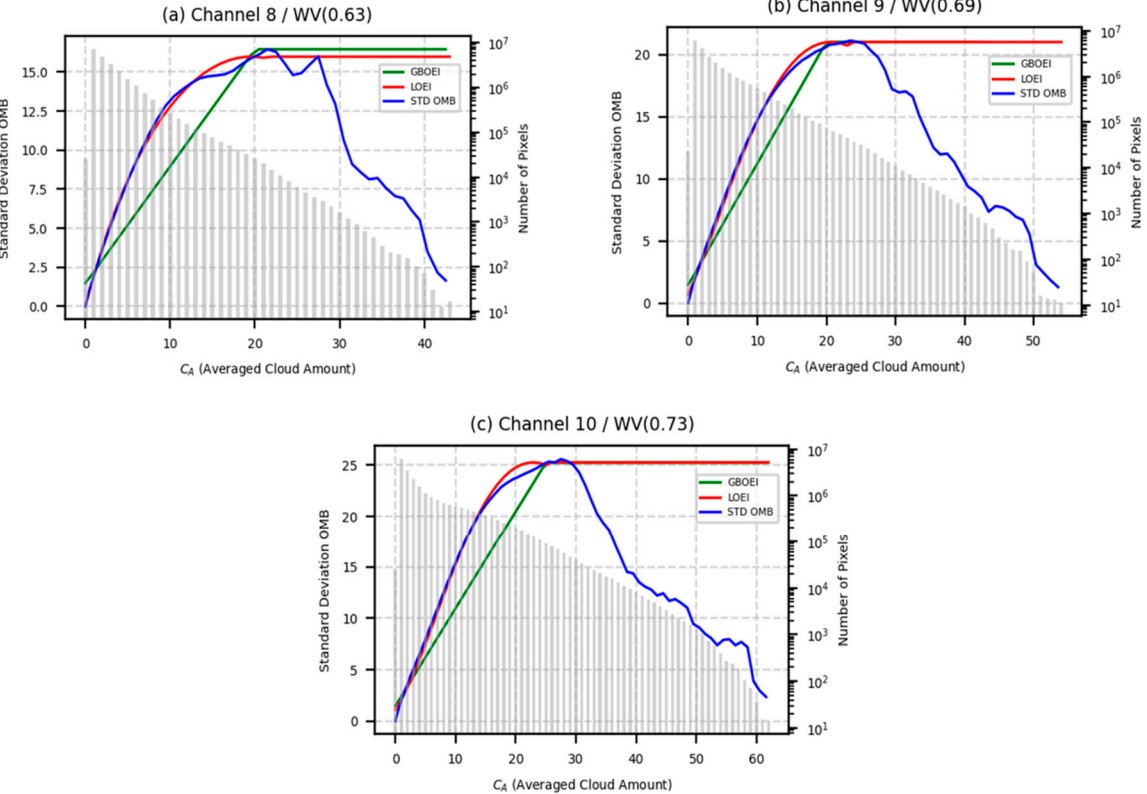

**Figure 6.** The standard deviation of OMB departures (blue lines) as a function of $C_A$ at the (**a**) channel 8, (**b**) channel 9, and (**c**) channel 10. The observation error estimated by GBOEI and LOEI are plotted by green and red lines, respectively. Grey bars are the log number of OMB samples without any QCs on the logarithmic scale at the right axis.

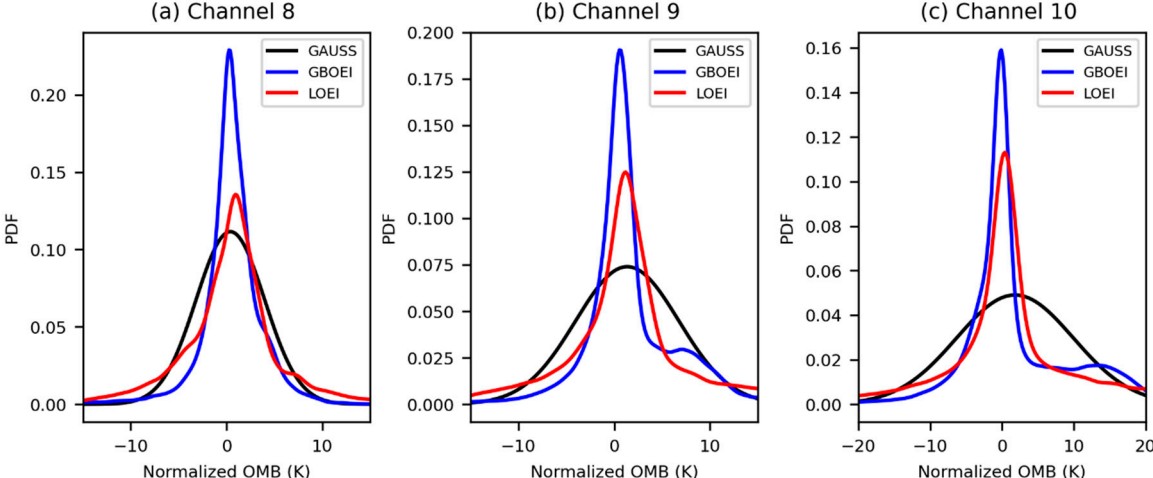

**Figure 7.** The PDFs of normalized OMB departures without any QCs at channel 8, 9, and 10. The normalization is calculated using standard deviation of GBOEI (blue lines) and LOEI (red lines). Black lines represent the Gaussian distribution.

### 3.2. Evaluation of Simulated BT Analysis

It is important to evaluate the effectiveness of GBOEI and LOEI in simulating BT in analysis fields. Figure 8 shows the density scatterplots of observation minus analysis (OMA) with the observation data using the samples from BT analysis at the last data assimilation time (2100 UTC 9 August 2020 and 1800 UTC 14 August 2020). We can see that the largest negative-tailed and asymmetric of OMA can be found in channel 10 compared

to other channels in both experiments. Channel 10 is sensitive to low tropospheric level, which means it is sensitive to clouds and precipitation. The negative-tails generally at the low observed BTs for all channels and experiments, while the positive-tails are relatively noticed in a broad of high BTs. This result indicates there was difficulty simulating the ASR in RTM probably due to the insufficient condensation cloud process for cloudy-skies (low BTs) and the excess absorption for the clear-skies (high BTs). However, the ExpLOEI experiment presented more symmetric pattern and smaller negative-tails, which indicates the BT analysis are closer to BT observation compared to ExpGBOEI experiment in all channels. It can be seen the error metric scores (MAE, STD, and RMSE) in LOEI experiment reduced by about $\pm 3.5\%$ for channel 8, $\pm 2\%$ for channel 9, and $\pm 30\%$ for channel 10, as compared to ExpGBOEI experiment. This result showed that the impact of ExpLOEI can be found primarily in channel 10.

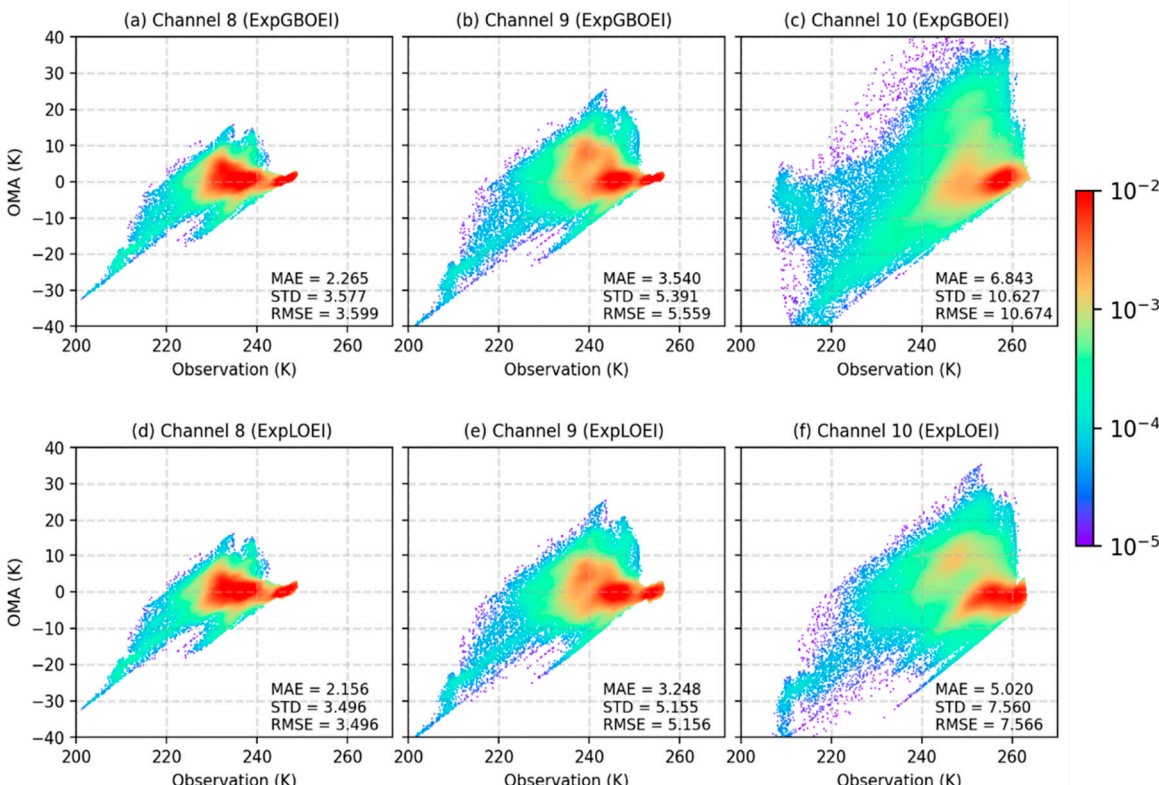

**Figure 8.** The density scatter plots and error statistics of OMA departures against BT observations at channel 8, 9, and 10 simulated by (**a**–**c**) ExpGBOEI and (**d**–**f**) ExpLOEI experiments.

Particularly, the BT analysis of channel 10 was classified into four cloud phases (clear, water, super-cooled, and ice) given in GK2A level-2 cloud phase product as shown in Figure 9. From the quantitative error (MAE, STD, and RMSE), it can be inferred that the ExpLOEI experiment showed positive impact in each individual cloud phases. However, the largest impact obviously found in super-cooled and ice phases, reaching at approximately $\pm 30\%$ of the average error (MAE, STD, and RMSE) statistic scores. This result proposes the cooler BT's associated with cloudy-sky are better represented in the ExpLOEI experiment.

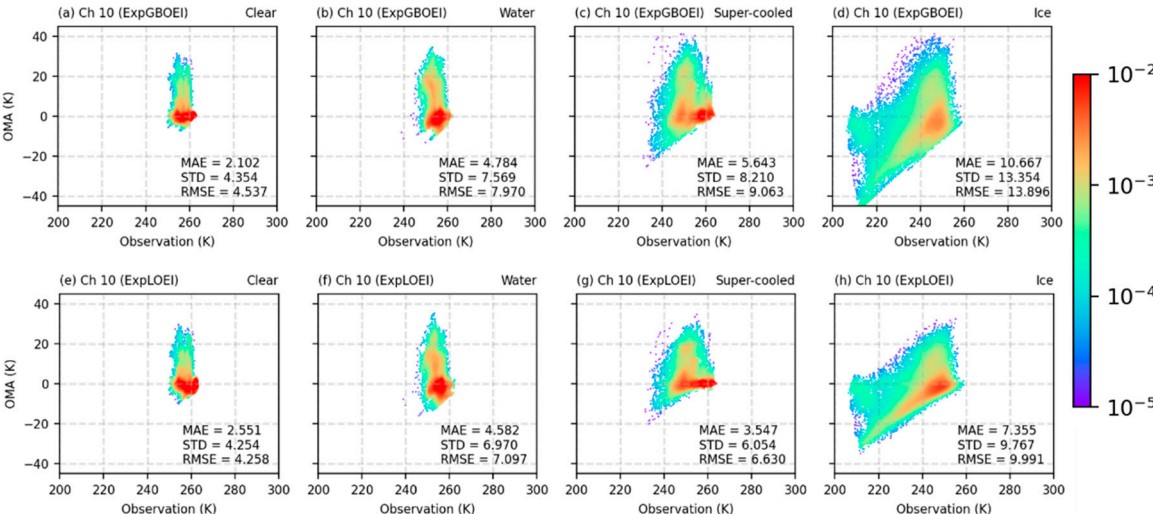

**Figure 9.** The density scatter plots and error statistics of OMA departures against BT observations at channel 10 simulated by (**a**–**d**) ExpGBOEI and (**e**–**h**) ExpLOEI experiments. The individual cloud phases is identified as (**a**,**e**) clear, (**b**,**f**) water, (**c**,**g**) super-cooled, and (**d**,**h**) ice given by GK2A AMI level-2 cloud phase product.

### 3.3. Increment of the Analysis Field

The numerical forecast model performance depends on the accuracy of the initial analysis field. The contribution of GBOEI and LOEI on improving the forecast models can be understand by analyzing the increment on the analysis field at the last cycling time. Figures 10 and 11 show the BT at channel 10, horizontal increment of mixing ratio and temperature at 850 hPa simulated by ExpGBOEI and ExpLOEI experiments. Figure 10 shows the results for 2100 UTC 9 August 2020 (Case 1). The convective elements were observed satellite BT observation in the east of Yellow Sea denoted by BT < 220 K. This convective element is associated with the moister atmospheric conditions. The water vapor mixing ratio in ExpGBOEI and ExpLOEI increased in the convective area where the low BT was located, with similar positive water vapor increments distribution. However, the amount of positive water vapor mixing ratio in ExpGBOEI was clearly larger than the ExpLOEI at approximately 2 g kg$^{-1}$. Furthermore, the drier region was observed by satellite observation denoted by BT > 207 K over the location A and B. Obviously, ExpGBOEI simulated a very strong negative water vapor mixing ratio over the location A and B at approximately higher than 5.0 g kg$^{-1}$, which would be unrealistic. In comparison, the weaker and narrower coverage of the negative water vapor mixing ratio are found in the ExpLOEI experiment. For the temperature increments, ExpGBOEI experiment showed weaker intensity in most of D02 coverage area compared to ExpLOEI experiment. Figure 11 shows the results for 1800 UTC 14 August 2020 (Case 2). The convective clouds was observed at approximately 36°N–122°E and 36.6°N–124°E which denoted by BT lower than 220 K over the Yellow Sea. Both ExpGBOEI and ExpLOEI increased the water vapor mixing ratio at approximately 4.5 g kg$^{-1}$ over the two location of convective area, with slightly different distribution. Moreover, over the location A and B, both ExpGBOEI and ExpLOEI experiments produced the negative water vapor increments in similar distribution. However, the ExpGBOEI simulated negative increments in tremendous intensity compared to the LOEI experiment. In temperature fields, the ExpGBOEI and ExpLOEI exhibited similar increments both in coverage and intensity.

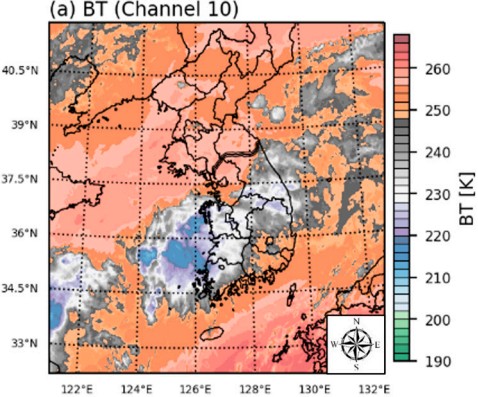

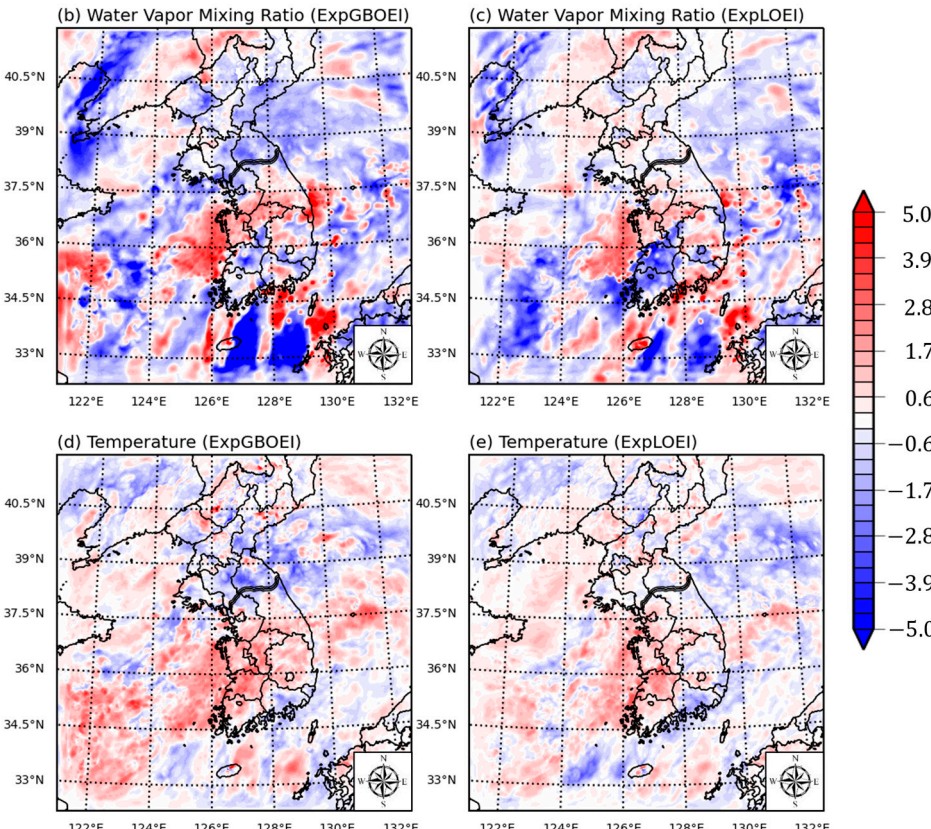

**Figure 10.** (**a**) The brightness temperature (K) at channel 10 and analysis horizontal increment of (**b**,**c**) water vapor mixing ratio (g kg$^{-1}$) and (**d**,**e**) temperature (K) simulated by (**b**,**d**) ExpGBOEI and (**c**,**e**) ExpLOEI experiments at 2100 UTC on 8 August 2020 (last cycling of data assimilation time) for Case 1.

Generally, comparing analysis increment between ExpGBOEI and ExpLOEI suggests that ExpLOEI produced more realistic water vapor mixing ratio and temperature increments compared to ExpGBOEI due to reduced BT differences between observation and analysis in the ExpLOEI experiment.

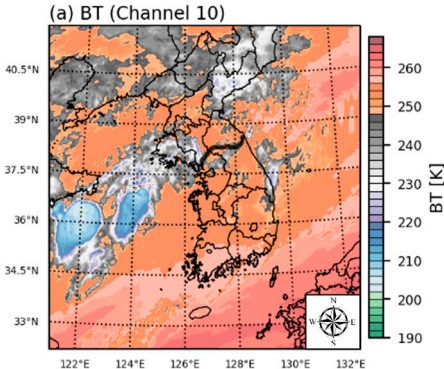

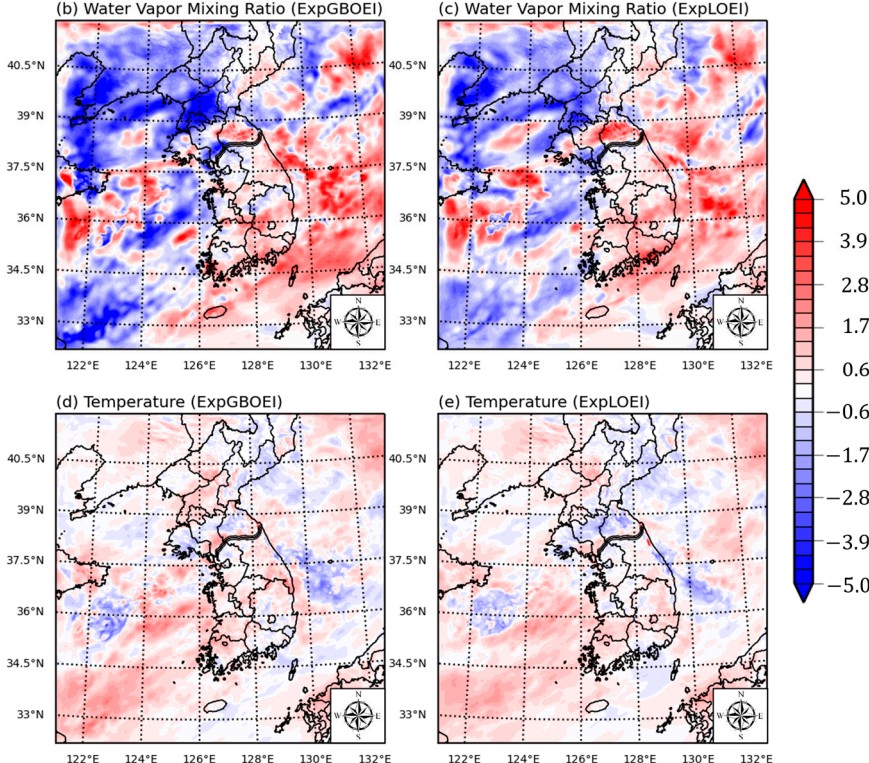

**Figure 11.** Same as Figure 10 except at 1800 UTC on 14 August 2020.

### 3.4. Distribution of the Cumulative Precipitation

The distribution of cumulative precipitation forecasts is displayed and analyzed with AWS observation data for Cases 1 and 2. Figure 12 show the 9-h cumulative rainfall from 2100 UTC on 8 August 2020 to 0600 UTC on 9 August 2020 generated by AWS, CTRL, ExpG-BOEI and ExpLOEI for Case 1. During 9 h period, an intense northeast-shifted rainband was observed over Gyeonggi Province. The two maximum precipitation areas marked by A and B was seen with the maximum precipitation reaching 120 mm. The CTRL experiment predicted small amount precipitation and broken northeast-shifted rainband, which missed the intense rainfall over locations A and B. The ExpGBOEI experiment generated an intense northeast-shifted rainband by more than 120 mm, clearly overestimated the AWS observations. This result caused by excessive positive water vapor mixing ratio in the analysis fields. In addition, the ExpGBOEI mislocated the intense rainfall in the north of location A, and underestimated the maximum center of location B for only 70 mm. In comparison, the ExpLOEI experiment exhibited a broader intense northeast-shifted rainband with the maximum rainfall reaching 120 mm which similar to observation. Moreover, the ExpLOEI

reduced the false alarms and well-matched the intense rainfall over the location A and slightly captured the maximum rainfall over location B.

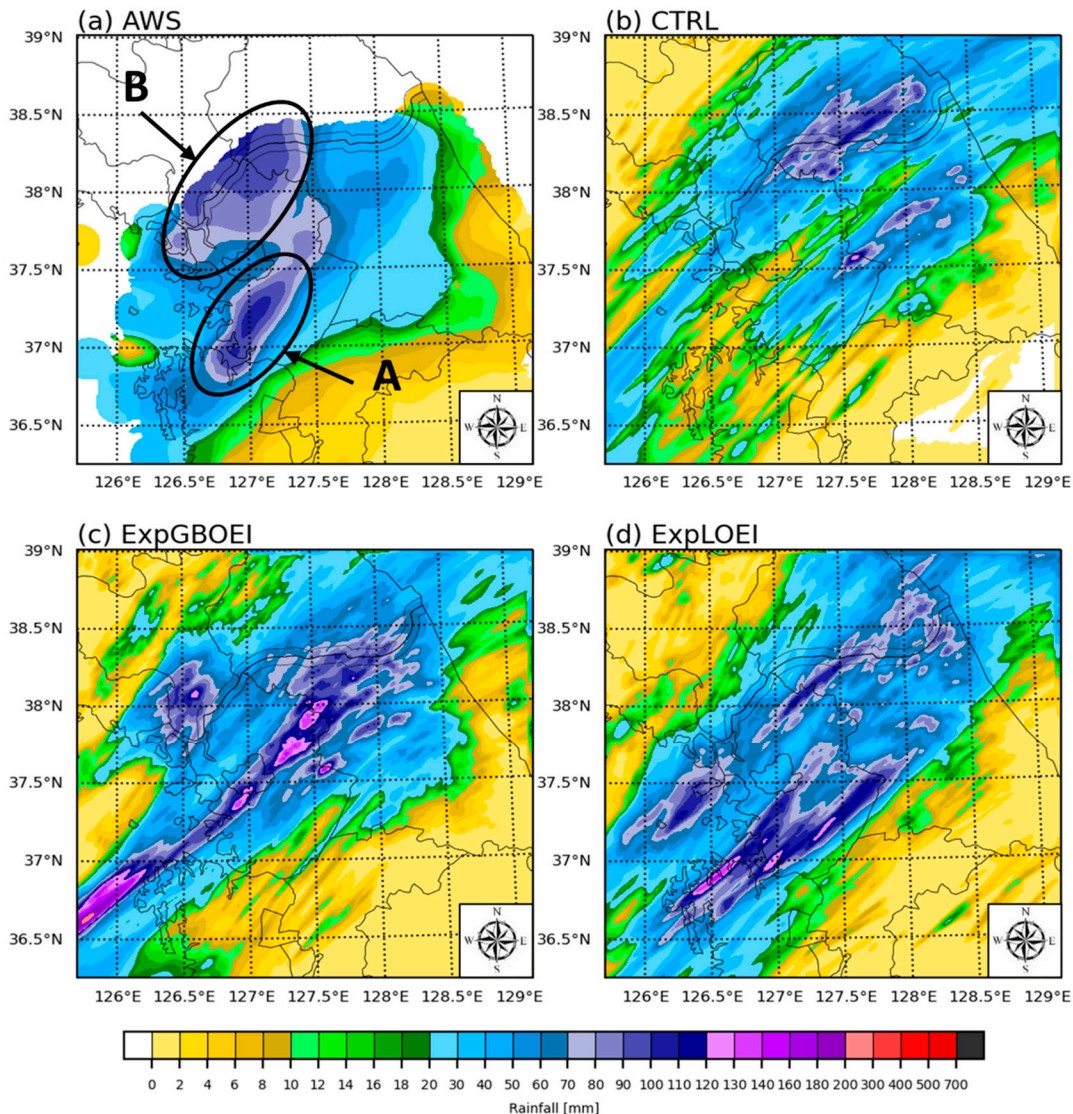

**Figure 12.** Cumulative precipitation (mm) distribution for Case 1 forecast period from the (**a**) AWS, (**b**) CTRL, (**c**) ExpGBOEI, and (**d**) ExpLOEI at domain 3. The marked A and B are the location of maximum precipitation.

Figure 13 show the 12-h cumulative precipitation from 1800 UTC on 14 August 2020 to 0600 UTC on 15 August 2020 of AWS, ExpGBOEI, and ExpLOEI, respectively. AWS observed an east-shifted rainband with the maximum precipitation center reaching 120 mm marked as location A. The CTRL experiment produced east-shifted rainband centered outside the inland of Korea Peninsula and missed the most location impact of heavy precipitation, as compared to AWS. The ExpGBOEI experiment also produced a weak and broken east-shifted rainband and substantially missed the intense rainfall over the location A. Meanwhile, the ExpLOEI experiment well-captured the east-shifted rainband compared to ExpGBOEI, which much similar pattern to the AWS observations. However, the ExpLOEI experiment still underpredicted the maximum precipitation in location A for only ±70 mm. Overall, the ExpLOEI experiment yields a better cumulative rainfall distribution in terms of coverage and intensity compared to ExpGBOEI and CTRL for Cases 1 and 2.

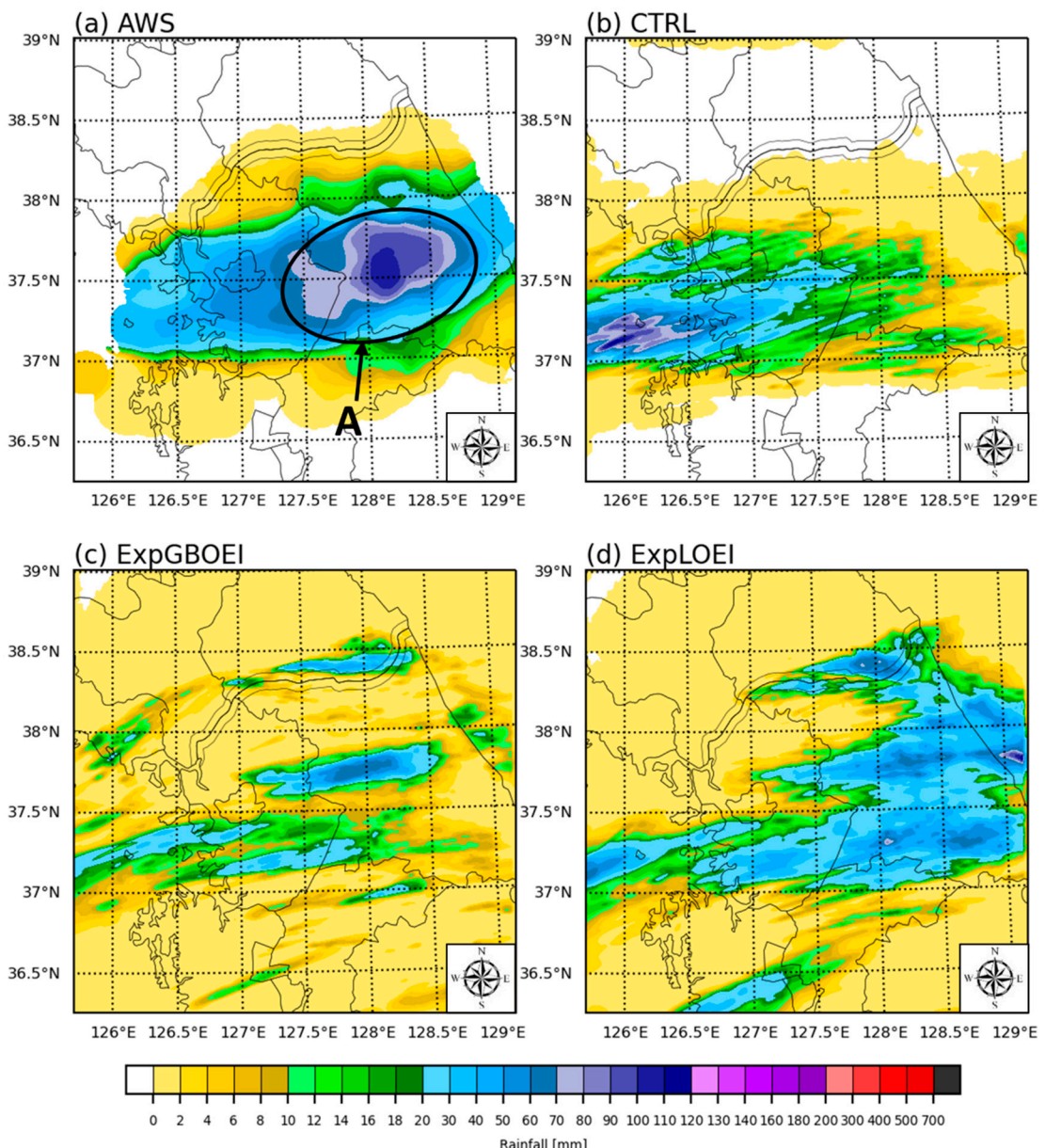

**Figure 13.** Same as Figure 12 except for Case 2 forecast period. The marked A is the location of maximum precipitation.

### 3.5. Model Verification

The cumulative precipitation of domain 3 was verified based on the quantitative error (RMSE and BIAS), categorical rainfall $\leq 0.1$ mm (AC and CSI), and PC against the AWS observations (Table 6). The influence distance of AWSs in South Korea has effectively 10 km $\times$ 10 km; thus, the precipitation models of domain 3 was averaged nine grid points when compared to AWSs point.

**Table 6.** Quantitative verification of cumulative precipitation for CTRL, ExpGBOEI, and ExpLOEI.

| | Experiment | RMSE (mm) | BIAS (mm) | AC | CSI | PC |
|---|---|---|---|---|---|---|
| | CTRL | 27.57 | −13.15 | 0.88 | 0.87 | 0.57 |
| Case 1 | ExpGBOEI | 27.01 | −2.44 | 0.91 | 0.90 | 0.63 |
| | ExpLOEI | 22.08 | 0.73 | 0.93 | 0.93 | 0.74 |
| | CTRL | 27.65 | −6.40 | 0.65 | 0.60 | 0.31 |
| Case 2 | ExpGBOEI | 26.21 | −13.87 | 0.72 | 0.69 | 0.53 |
| | ExpLOEI | 22.51 | −6.22 | 0.79 | 0.76 | 0.55 |
| | CTRL | 27.61 | −9.77 | 0.76 | 0.73 | 0.44 |
| Average | ExpGBOEI | 26.61 | −8.15 | 0.81 | 0.79 | 0.58 |
| | ExpLOEI | 22.29 | −2.74 | 0.86 | 0.84 | 0.64 |

For the two cases (Case 1 and 2), the CTRL underestimated precipitation in both location and intensity which produced negative BIAS. The ExpGBOEI experiment increased the precipitation intensity and reduced the negative BIAS of CTRL in Case 1, but reduced the intensity of rainfall and increased the negative BIAS in Case 2. While, the ExpLOEI experiment reduced the negative BIAS of CTRL and showed the lowest BIAS for all cases. For RMSE, the ExpGBOEI and ExpLOEI experiment were reduced at approximately 0.6 mm and 5 mm, respectively, as compared to CTRL for all cases. Considering the category classification, the scores of all experiments were mostly comparable to CTRL. However, both ASR assimilation experiments produced better performances than those of the CTRL, with the best performances of AC and CSI scores was achieved by ExpLOEI experiment. Similarly, PC scores show the highest scores was produced by the ExpLOEI experiment in both cases, as compared to all experiments. The result of averaging all of the metrics scores show the ExpLOEI experiment has the best performance. The reduction error rates of RMSE and BIAS in ExpLOEI experiment reached ±5 mm and ±7 mm compared to CTRL. In categorical quantitative, the ExpLOEI experiment showed the value of percentage increase of AC and CSI (i.e., 11% for the AC and 13% for the CSI) as compared to CTRL. Considering the pattern correlation, the ExpLOEI increased the scores at approximately 31d while the ExpGBOEI's PC scores improved only ±24%.

The precipitation models were also verified as a function of lead times. Figures 14 and 15 show the quantitative error (RMSE and BIAS) and categorical rainfall ≤ 0.1 mm (AC and CSI) calculated using the hourly forecast models and AWS for Case 1 and 2, respectively. The CTRL experiment underpredicted the precipitation throughout the forecast times as shown by negative BIAS. Hence, the AC and CSI scores were the lowest relative to other experiments for most of the forecast times in both Cases. The ExpGBOEI and ExpLOEI reduced the negative BIAS of CTRL due to increased precipitation in both Cases. However, there was a strong overestimation (BIAS ≥ 2 mm) in ExpGBOEI and ExpLOEI experiments at the 3rd and 4th h (Case 1) for the ExpGBOEI experiment and at the first 2 h (Case 2) for the ExpLOEI experiment. For RMSE, the ExpGBOEI experiment showed a variative impact and only small reduction of RMSE at most of prediction hour in both cases, as compared to CTRL. Meanwhile, the ExpLOEI experiment reduced the RMSE error at the entire of forecast range in both cases, but slightly increased error at the first h in Case 2. In categorical validation (AC and CSI), it can be inferred the positive impact of ExpGBOEI and ExpLOEI experiments occurred at between 3rd and 8th h for Case 1 and the first 5 h for Case 2. However, the ExpLOEI experiment was evidently better than the ExpGBOEI experiment, leading to 28% and 50% (Case 1) and 25% and 45% (Case 2) improvements in AC and CSI, respectively, as compared to CTRL. Overall, the ExpLOEI experiment eventually leads to a better performance of the quantitative precipitation forecasts against the AWSs data for most forecast times.

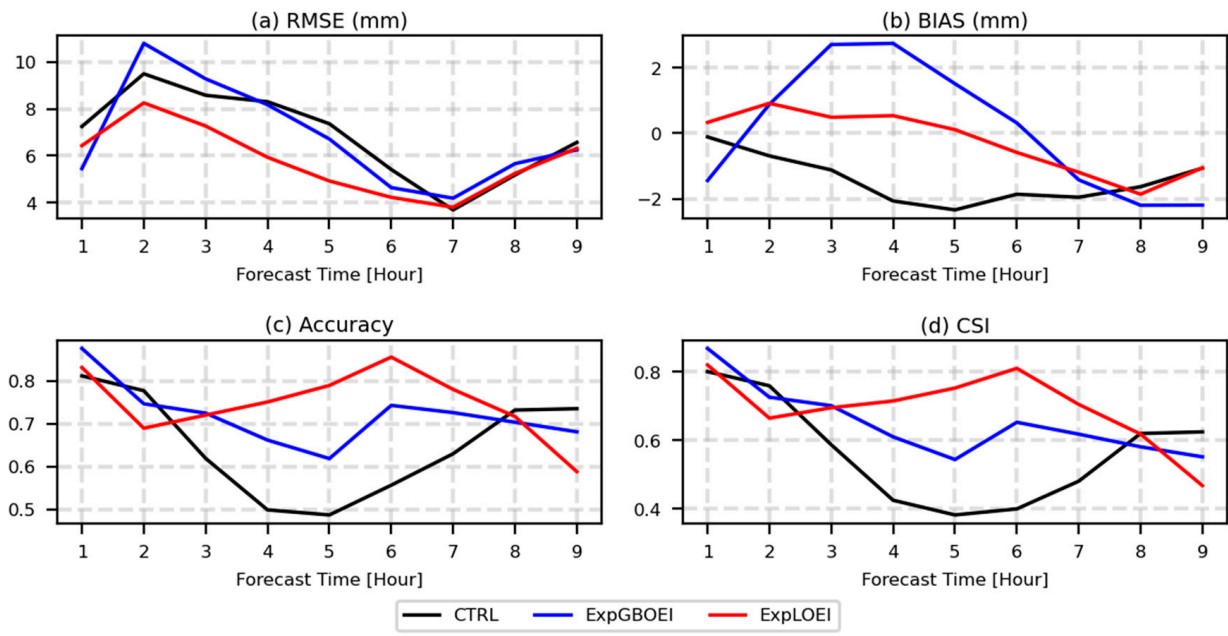

**Figure 14.** Hourly verification statistics of the (**a**) root mean square error (RMSE), (**b**) BIAS, (**c**) Accuracy (AC) and, (**d**) critical success index (CSI) for CTRL (black lines), ExpGBOEI (blue lines), and ExpLOEI (red lines).

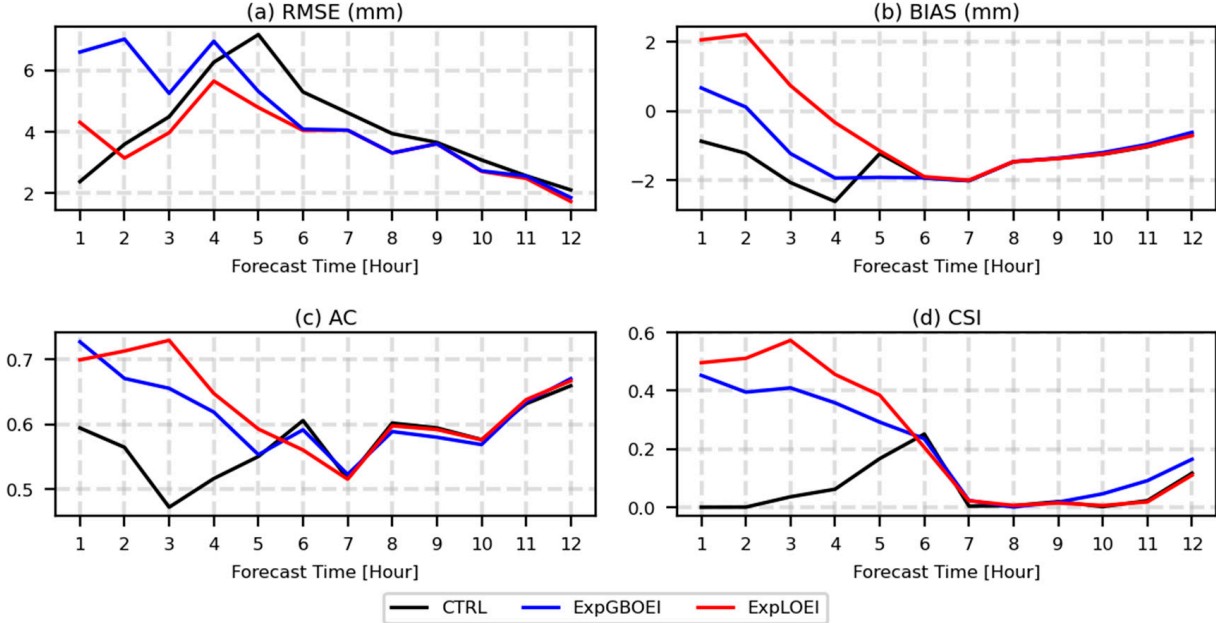

**Figure 15.** Same as Figure 14 except for Case 2.

More specifically, the calculation of ETS was performed using the hourly precipitation forecast and AWSs data based on four hourly rainfall thresholds, i.e., light rain ($0.1 \leq R \leq 3$), moderate rain ($3 \leq R \leq 15$), heavy rain ($15 \leq R \leq 30$), and very heavy rain ($R \geq 30$) (Figure 16). The averaged of hourly ETS was calculated for both Cases. Afterwards, the hourly ETS of two Cases was also averaged. From Figure 16, it can be seen that both ExpGBOEI and ExpLOEI experiments improved ETS for all rainfall thresholds compared to CTRL experiment. The ExpLOEI experiment performed better than the ExpGBOEI experiment in all rainfall thresholds (except for the very heavy rain threshold). Compared to ExpGBOEI, the ExpLOEI experiment exhibited $\pm 27\%$ ETS improvement percentage

for light rain, ±14% improvement percentage for moderate rain, and ±37% improvement percentage for heavy rain. Meanwhile, in very heavy rainfall threshold, a ±36% ETS percentage decrease was found in the ExpLOEI experiment as compared to the ExpGBOEI experiment.

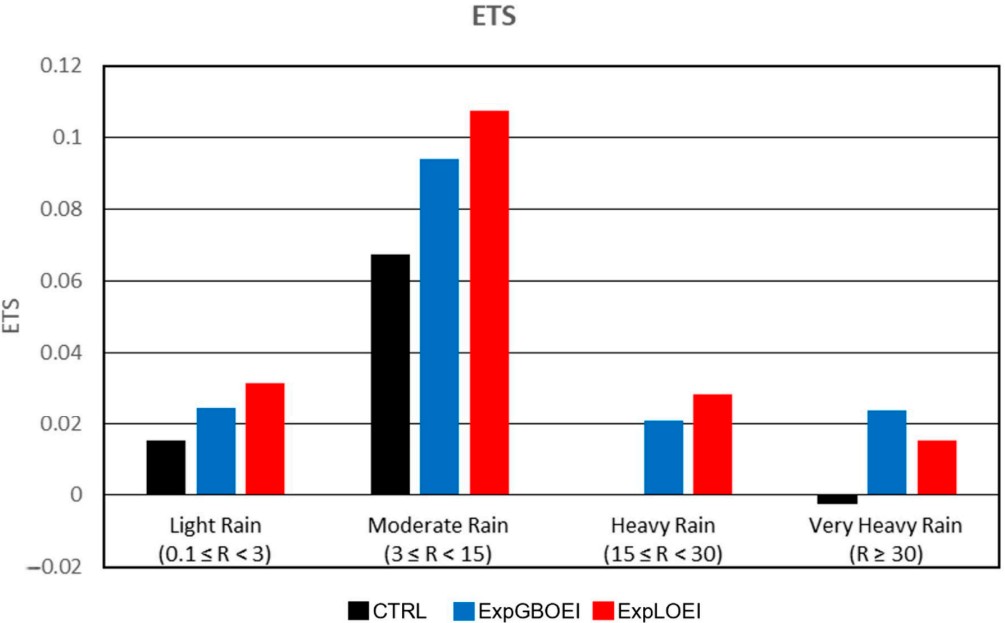

**Figure 16.** Average of hourly ETS verification statistics of Case 1 and 2 simulated by CTRL (black bars), ExpGBOEI (blue bars), and ExpLOEI (red bars) experiments.

The assimilation of ASR is most sensitive to moisture and thermodynamic fields in numerical models. The vertical profile of the water vapor mixing ratio and temperature fields was verified by using the Osan radiosondes at the 3 h forecast for Case 1 and the 4 h forecast for Case 2 (Figure 17). The water vapor mixing ratio was underestimated by CTRL in both Cases at the most of pressure levels. In both cases, the ExpGBOEI experiment reduced the water vapor mixing ratio underestimation at the most of pressure levels, except producing even worser underestimation of water vapor mixing ratio from surface to 400 hPa in Case 2, as compared to CTRL. However, the ExpLOEI experiment improved the water vapor mixing ratio vertical profile at the entire of pressure levels without any exception compared to CTRL. Similarly, temperature profiles biases are shown in Figure 17b,d. It can be inferred that the impact of ASR in both experiments was more variative in temperature profiles than in the water vapor mixing ratio (which had either a positive and negative impact according to atmospheric levels). However, at the most of atmospheric layers, both ExpGBOEI and ExpLOEI experiments reduced the vertical biases compared to CTRL. The lowest biases frequently produced by the ExpLOEI experiment in all layers (especially near the surface levels due to BT analysis improvements at channel 10, which is more sensitive at the lower level).

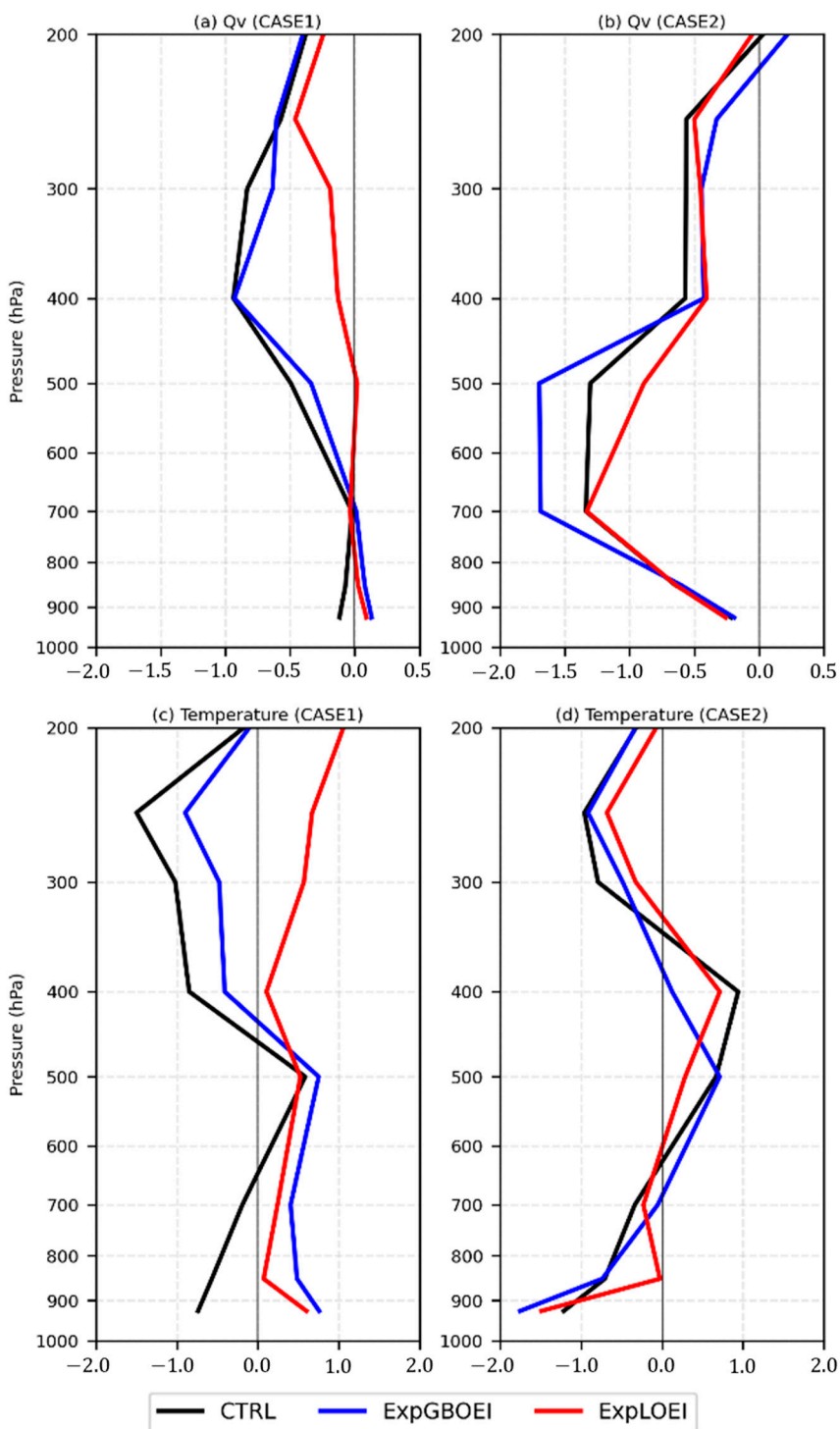

**Figure 17.** Vertical profiles biases of (**a**,**b**) water vapor mixing ratio (g kg$^{-1}$) and (**c**,**d**) temperature (°C) at (**a**,**c**) 2020.08.09.0000 UTC (Case 1) and (**b**,**d**) 2020.08.15.0000 UTC (Case 2) for CTRL (black lines), ExpGBOEI (blue lines), and ExpLOEI (red lines) experiments.

## 4. Conclusions

In this study, the LOEI is introduced for assimilating ASR on the 3DVAR data assimilation system. The ASR data were obtained from the three water vapor channels of the new generation geostationary satellite GK-2A. The LOEI were estimated using the $C_A$ and STD of OMB samples in August 2020 every six h without QCs. The experiments were conducted for the simulation of two summertime precipitation in South Korea using the

high-resolution WRF model. The assimilation of ASR with LOEI and GBOEI, referred as ExpLOEI and ExpGBOEI experiments, respectively, were studied and compared. Also, as the measure of the effectiveness of ExpGBOEI and ExpLOEI, simulation without data assimilation (i.e., the CTRL experiment) were presented. The simulation results were analyzed thoroughly by displaying the PDFs of normalized OMB, analysis of OMA departures, accumulated precipitation, and verification statistics.

The comparison between GBOEI and LOEI showed that the LOEI better fit the real-value of STD of OMB in the three water vapor channels compared to GBOEI. The LOEI also did not require threshold parameters which alleviates the uncertainties. The PDFs of OMB normalized by LOEI formed a closer distribution to Gaussian than normalized by GBOEI in all water vapor channels, which can assimilate successfully ASR into data assimilation system. This result contributed to the better BT analysis in ExpLOEI experiment which indicated by the reduction of fat-tails distribution and error statistics in the analysis of OMA. The improvement of BT analysis primary can be found in channel 10 since this channel is more sensitive to clouds than any other humidity channels. In specific of channel 10, the ExpLOEI experiment mainly reduced the error statistics of OMA in supercooled and ice sky phases compared to clear and water phases. This result suggested that LOEI were able to treat correctly radiances at the large $C_A$. This improvement of BT analysis in ExpLOEI experiment eventually provided more accurate variables data in WRF model. The increment of water vapor mixing ratio and temperature analysis fields in ExpLOEI experiment showed a reasonable intensity which primary has smaller increment intensity than the ExpGBOEI experiment in two cases. Due to the modification of variable analysis, the cumulative precipitation in ExpLOEI experiment has better agreement than ExpGBOEI experiment in two cases. From the verification statistics, the ExpLOEI experiment decreased the error statistics and increased the categorical scores compared to ExpGBOEI and CTRL. It was found that the ExpLOEI experiment obviously has positive impact on the forecast lead times, even though the 1–2nd h of forecast the statistics score has both negative and positive impact. In rainfall threshold statistics, both ExpLOEI and ExpGBOEI experiments increased the ETS score for light, moderate, heavy, and very heavy rain thresholds. However, the ExpLOEI experiment has higher score than ExpGBOEI experiments, except for very heavy rain. Furthermore, the forecast variable verification statistics showed that the ExpLOEI experiment mostly improved the water vapor mixing ratio at the lower level. This improvement may be caused by the large improvement of BT analysis in channel 10, which is more sensitive to lower levels than other water vapor channels.

In this study, the 3DVAR technique was used for assimilation. However, 3DVAR is limited by several issues such as the use of static background error covariances, making it impractical for simulating the flow-dependent feature of the atmosphere. The ensemble-based assimilation method is expected to provide flow-dependent background error covariances. In addition, only water vapor channels were used to assimilate the ASR. Assimilating other surface infrared channels will pose various challenges and requires a careful treatment of surface emissivity.

**Author Contributions:** M.I.H. wrote the first draft, assisted with data curation, and performed formal analysis, investigation, software coding, and visualization of the research article. K.-H.M. conceptualized the paper, supervised and administered the project, performed formal analysis, provided resources and funding, and reviewed and edited the paper. All authors have read and agreed to the published version of the manuscript.

**Funding:** This work is funded by the Korea Meteorological Administration Research and Development Program under grant KMI2022-00310 (NARAE-Weather) and grant KMA2018-00125 (Observing Severe Weather in Seoul Metropolitan Area and Developing Its Application Technology for Forecasts).

**Conflicts of Interest:** The authors declare no conflict of interest. The funders had no role in the design of the study; in the collection, analysis, or interpretation of data; in the writing of the manuscript; or in the decision to publish the results.

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
