# Peer review of "Impact of Assimilating GK-2A All-Sky Radiance with a New Observation Error for Summer Precipitation Forecasting"

_remotesensing, doi:10.3390/rs15123113_

Round 1

Reviewer 1 Report

The manuscript entitled "Impact of Assimilating GK-2A All-Sky Radiance with a New Observation Error for Summer Precipitation Forecasting" can be used as a reference for future studies. However, the authors must address the following issues to make it suitable for further revision.

Ü  The manuscript has numerous grammatical and methodological issues that need to be checked, and it needs to restructure the contents to improve clarity, content, and organization. Thus, these concerns must be appropriately addressed and then reviewed again.

Ü  According to MDPI/ any other reputable journals rule, your manuscript should have line numbers displayed, which allows you to reference exactly where a particular error is and that will be able to easily locate any problem areas.

Ü  The manuscript's marginal scientific contribution/ nobility is unclear in the abstract or the introduction. What is the marginal scientific contribution/ nobility of your manuscript?

Ü  Literature on a very similar topic that has already been published ( e.g., https://doi.org/10.1007/s00376-020-0219-z, https://doi.org/10.3390/rs15112760, https://doi.org/10.3390/rs15112760 or others) should be included in your review may encourage you to take a new approach to the subject that has never been covered before.  

Ü  Figure 4 misses the basic cartographic elements like the grid, north arrow, and other related ones. Please, try to be consistent as you did in Figure 9; however, it also needs revision.

Extensive editing of English language required

Author Response

We appreciate the reviewer for your precious time in reviewing our paper and providing valuable comments. It was your valuable and insightful comments that led to possible improvements in the current version. The authors have carefully considered the comments and tried our best to address every one of them.

Point 1.  The manuscript has numerous grammatical and methodological issues that need to be checked, and it needs to restructure the contents to improve clarity, content, and organization. Thus, these concerns must be appropriately addressed and then reviewed again.

Response 1. We had restructured and revised grammatical and methodological issues. Please refer to Author track change’s file.

Point 2. According to MDPI/ any other reputable journals rule, your manuscript should have line numbers displayed, which allows you to reference exactly where a particular error is and that will be able to easily locate any problem areas.

Response 2. We had displayed the line numbers. Please refer to Author track change’s file.

Point 3. The manuscript's marginal scientific contribution/ nobility is unclear in the abstract or the introduction. What is the marginal scientific contribution/ nobility of your manuscript?

Response 3. The abstract and introduction has been reconstructed and revised. Please refer to Author track change’s file. This marginal scientific contribution of this study is introducing a new observation error.

Point 4.  Literature on a very similar topic that has already been published ( e.g., https://doi.org/10.1007/s00376-020-0219-z, https://doi.org/10.3390/rs15112760, https://doi.org/10.3390/rs15112760 or others) should be included in your review may encourage you to take a new approach to the subject that has never been covered before. 

Response 4. The literature of e.g., https://doi.org/10.1007/s00376-020-0219-z has different observation error method with our study. They used observation error from Geer-Bauer.

Point 5.  Figure 4 misses the basic cartographic elements like the grid, north arrow, and other related ones. Please, try to be consistent as you did in Figure 9; however, it also needs revision.

Response 5. The basic cartographic elements has been added in figure 4. Please refer to Author track change’s file.

Comments on English

Point 1. Extensive editing of English language required

Response 1. We had re-read and revised the english languange of our manuscript. Please refer to Author track change’s file.

Reviewer 2 Report

This manuscript proposes an innovative method of data assimilation for precipitation forecast. The proposed method shows higher accuracy in precipitation location and intensity compared to the traditional linier function method. Here are some suggestions for further improving the quality of this manuscript.

1. There are some typographical errors in the manuscript, such as Inconsistent thickness of table borders, variables not italicized, and inconsistent indentation at the beginning of sentences. Carefully check.

2. There are some grammatical errors in the manuscript, such as singular and plural forms of verbs, missing comma at the end of the sentence, and so on. Carefully check and correct them.

3. The same abbreviation and its full name are used multiple times in different locations. check and correct them.

4. The variables in the Equations should be in italics.

5. Introduction Section, the second sentence. Some very relevant research work on weather prediction using different types of observation data was omitted. It should be considered in the literature review. For example,

[1] Sibolla, B.H., Van Zyl, T. & Coetzee, S. Determining Real-Time Patterns of Lightning Strikes from Sensor Observations. J geovis spat anal 5, 4 (2021). https://doi.org/10.1007/s41651-020-00070-7

[2] Ahmed, A., Nawaz, R., Woulds, C. et al. Influence of hydro-climatic factors on future coastal land susceptibility to erosion in Bangladesh: a geospatial modelling approach. J geovis spat anal 4, 6 (2020). https://doi.org/10.1007/s41651-020-00050-x

6. Regarding the sixth paragraph of the Introduction section, have WRF and 3DVAR been used in all-sky radiance assimilation methods prior to this manuscript's proposed method? Or are they exclusively used in this new approach? Additionally, what is the rationale behind employing WRF and 3DVAR in this manuscript's method?

7. This manuscript focuses on introducing a Model, but Section 2 delves into conventional knowledge that could be detracting from the main purpose. It may be beneficial to refine and modify the manuscript to ensure that the proposed model is clearly and prominently introduced.

8. Add a flowchart of the proposed model.

9. Check the sequence numbers of the subsection headings in the Results section.

10. The Summary and Conclusion section discusses the NEW_OE model, yet it is not evident from the abstract and title. What makes this model innovative? The abstract and title contain references to "new observation error" and "pre-calculated radiance error." Are these related to the use of the look-up table in the observation error?

11. Table 1. Make a triple line border table. Besides, note that the title of the table already includes GK-2A AMI, hence repetition in the table header is unnecessary. Finally, ensure that the table's bottom border is made bold.

12. To ensure consistency throughout the manuscript, the abbreviation used to refer to figures or diagrams should be uniform, whether it is expressed as Fig. or Figure

13. The caption for Figure 1 includes the full name of AWS in the previous text, so there is no need for it to be repeated here. Revise the sentence accordingly.

14. Regarding the first sentence and the 2.2 title which references WRFDA and WRF 3D-Var, respectively, could you clarify if these are referring to the same model? The different forms of expression make it challenging to comprehend.

15. Why use the simple linear function proposed by Geer and Bauer for comparison? Why not use other methods?

16. Figures 4(b) and 4(e) are too blurry to see clearly.

17. The clarity of Figures 9a and 10a is insufficient to view them properly.

18. What does the y-axis represent in Figure 15?

19. Why was the 3DVAR method used instead of the ensemble-based assimilation method in the manuscript? What was the reasoning behind this decision?

Minor editing of English languate required

Author Response

We appreciate the reviewer for your precious time in reviewing our paper and providing valuable comments. It was your valuable and insightful comments that led to possible improvements in the current version. The authors have carefully considered the comments and tried our best to address every one of them.

Point 1. There are some typographical errors in the manuscript, such as Inconsistent thickness of table borders, variables not italicized, and inconsistent indentation at the beginning of sentences. Carefully check.

Response 1. The typographical errors has been revised. Please refer to Author track change’s file.

Point 2. There are some grammatical errors in the manuscript, such as singular and plural forms of verbs, missing comma at the end of the sentence, and so on. Carefully check and correct them.

Response 2. The grammatical errors has been revised. Please refer to Author track change’s file.

Point 3. The same abbreviation and its full name are used multiple times in different locations. check and correct them.

Response 3. We have revised this issues. Please refer to Author track change’s file

Point 4. The variables in the Equations should be in italics.

Response 4. We have revised this issues. Please refer to Author track change’s file

Point 5. Introduction Section, the second sentence. Some very relevant research work on weather prediction using different types of observation data was omitted. It should be considered in the literature review. For example,

[1] Sibolla, B.H., Van Zyl, T. & Coetzee, S. Determining Real-Time Patterns of Lightning Strikes from Sensor Observations. J geovis spat anal 5, 4 (2021). https://doi.org/10.1007/s41651-020-00070-7

[2] Ahmed, A., Nawaz, R., Woulds, C. et al. Influence of hydro-climatic factors on future coastal land susceptibility to erosion in Bangladesh: a geospatial modelling approach. J geovis spat anal 4, 6 (2020). https://doi.org/10.1007/s41651-020-00050-x

Response 5. We added the literatures. Please refer to Author track change’s file.

Point 6. Regarding the sixth paragraph of the Introduction section, have WRF and 3DVAR been used in all-sky radiance assimilation methods prior to this manuscript's proposed method? Or are they exclusively used in this new approach? Additionally, what is the rationale behind employing WRF and 3DVAR in this manuscript's method?

Response 6. This novel of this study is introducing a new observation error, and we investigated them into 3DVAR data assimilation from WRF model. 3DVAR from WRF model is used because it has low-computational cost and we want to do the first testing of our observation error.

Point 7. This manuscript focuses on introducing a Model, but Section 2 delves into conventional knowledge that could be detracting from the main purpose. It may be beneficial to refine and modify the manuscript to ensure that the proposed model is clearly and prominently introduced.

Response 7. We have revised section 2 and focused on introducing new observation error. Please refer to Author track change’s file.

Point 8. Add a flowchart of the proposed model.

Response 8. We added Flowchart as Figure 2. Please refer to Author track change’s file.

Point 9. Check the sequence numbers of the subsection headings in the Results section.

Response 9. We have revised the subsection headings. Please refer to Author track change’s file.

Point 10. The Summary and Conclusion section discusses the NEW_OE model, yet it is not evident from the abstract and title. What makes this model innovative? The abstract and title contain references to "new observation error" and "pre-calculated radiance error." Are these related to the use of the look-up table in the observation error?

Response 10. We have revised this, and we chose one abbreviation for NEW_OE, new observation error, and "pre-calculated radiance error as LOEI. Please refer to Author track change’s file.

Point 11. Table 1. Make a triple line border table. Besides, note that the title of the table already includes GK-2A AMI, hence repetition in the table header is unnecessary. Finally, ensure that the table's bottom border is made bold.

Response 11. We have revised this table. Please refer to Author track change’s file.

Point 12. To ensure consistency throughout the manuscript, the abbreviation used to refer to figures or diagrams should be uniform, whether it is expressed as Fig. or Figure

Response 12. The abbreviation in the entire manuscript and figures has been revised.

Point 13. The caption for Figure 1 includes the full name of AWS in the previous text, so there is no need for it to be repeated here. Revise the sentence accordingly.

Response 13. We have revised the figure 1. Please refer to Author track change’s file.

Point 14. Regarding the first sentence and the 2.2 title which references WRFDA and WRF 3D-Var, respectively, could you clarify if these are referring to the same model? The different forms of expression make it challenging to comprehend.

Response 14. We have chosen one expression for this (WRF, WRFDA, and 3DVAR) and changed the entire manuscript. Please refer to Author track change’s file.

Point 15. Why use the simple linear function proposed by Geer and Bauer for comparison? Why not use other methods?

Response 15. This is because Geer and Bauer method has extensively used in many research, and this study is one of pioneer in improving observation error model for all-sky radiance. Please refer to Author track change’s file on introduction line 67-88.

Point 16. Figures 4(b) and 4(e) are too blurry to see clearly.

Response 16. We had increased the Figures 4(b) and 4(e) quality as much as possible.  Please refer to Author track change’s file in figure 5(b) and 5(e).

Point 17. The clarity of Figures 9a and 10a is insufficient to view them properly.

Response 17. We believe that the clarity of Figures 9a and 10a is proper. Please refer to Author track change’s file in figure 10(a) and 11(a).

Point 18. What does the y-axis represent in Figure 15?

Response 18. We presented the y-axis in Figure 15. Please refer to Author track change’s file in figure 16.

Point 19. Why was the 3DVAR method used instead of the ensemble-based assimilation method in the manuscript? What was the reasoning behind this decision?

Response 19. This is the first attempt of assimilating all-sky radiance from GK-2A with a new observation error. This error is a look-up-table approach which can used directly to 3DVAR, so we decided to investigated this error in 3DVAR framework. The future study would be using ensemble method.

Comments on the Quality of English Language

Point 1. Minor editing of English languate required

Response 1. We re-read and revised the english languange of our manusript. Please refer to Author track change’s file

Reviewer 3 Report

The research entitled " Impact of assimilating GK-2A All-Sky Radiance with a New Observation Error for Summer Precipitation Forecasting" presents a good approach to improve the all-sky radiance assimilation of a Korean second geostationary meteorological satellite. The research design is strong, analyzed and presented well. It can be accepted after minor recheck.

English quality of the paper is satisfactory. However, a minor spell and grammatical check should be done. 

Author Response

We appreciate the reviewer for your precious time in reviewing our paper and providing valuable comments. It was your valuable and insightful comments that led to possible improvements in the current version. The authors have carefully considered the comments and tried our best to address every one of them.

Reviewer 4 Report

Thank you for submitting your paper to Journal of Remote Sensing. I read carefully manuscript number: remotesensing-2389471. The paper entitled " Impact of Assimilating GK-2A All-Sky Radiance with a New Observation Error for Summer Precipitation Forecasting" is well organized and scientific contents of the manuscript is of great interest for readers of Remote Sensing. This study introduces a newly observation error for assimilating all-sky radiances from GEO-KOMPSAT-2A (GK-2A) geostationary satellite. The paper needs minor modifications before it is processed:

1- The north arrow and scale bar should be added to Figures 1, 2, 4, 9, 10 and 11.

2- Figure legend should be added to Figures 10 and 11.

3- The "Summary and Conclusions" section need to be rewritten.

4- Figure legend should be added to Figure 16.

5-  Eq. 11-15 needs references.

According the above-mentioned comments, the paper requires minor revisions before publication. Thank you again for your submission, and I look forward to seeing the revised version of your manuscript.

Minor editing of English language required

Author Response

We appreciate the reviewer for your precious time in reviewing our paper and providing valuable comments. It was your valuable and insightful comments that led to possible improvements in the current version. The authors have carefully considered the comments and tried our best to address every one of them.

Point 1. The north arrow and scale bar should be added to Figures 1, 2, 4, 9, 10 and 11.

Response 1. We added the arrow and scale bar to Figures 1, 2, 4, 9, 10 and 11. Please refer to Author track change’s file in Figure 1, 3, 5, 10, 11, 12, 13.

Point 2. Figure legend should be added to Figures 10 and 11.

Response 2. We added the arrow and scale bar to Figures 10 and 11. Please refer to Author track change’s file in Figure 12 and 13.

Point 3. The "Summary and Conclusions" section need to be rewritten.

Response 3. We rewrote the conclusions. Please refer to Author track change’s file.

Point 4. Figure legend should be added to Figure 16.

Response 4. We added legend on Figure 16. Please refer to Author track change’s file in Figure 17

Point 5.  Eq. 11-15 needs references.

Response 5. We added references on Eq.11-15. Please refer to Author track change’s file.

Comments on the Quality of English Language

Point 1. Minor editing of English language required

Response 1. We had read the entire manuscript and revised the english languange again. Please refer to the author track change’s file

Round 2

Reviewer 1 Report

The authors improved the revised version. Still, the following issues must be addressed to make it suitable for publication.

Figure 5 should be revised and consistent with Figure 9 per the earlier comment.

Ü  The legend is merged with the maps that should be separated.

Ü  It misses the basic cartographic elements like the grid, north arrow, and other related ones that need revision.

Ü  The letters ( a-f) and numbers( 10)should not hide the pieces of information on the map. So, make it transparent for the background map information.  

Ü  What is the meaning of  #10 and   other symbols? Please, avoid such confusion for your potential readers.

In Figure 10-13, the north arrow  should be placed on the maps per the cartographic/map-making/ principles, not outside the map.  

Moderate editing of English language required

Author Response

We appreciate the reviewer for your precious time in reviewing our paper for second round and providing valuable comments. It was your valuable and insightful comments that led to possible improvements in the current version. The authors have carefully considered the comments and tried our best to address every one of them.

Figure 5 should be revised and consistent with Figure 9 per the earlier comment.

Point 1.   The legend is merged with the maps that should be separated.

Response 1. the legend has been seperated, Please refer to the author's track change file

Point 2.   It misses the basic cartographic elements like the grid, north arrow, and other related ones that need revision.

Response 2. The grid north, legend wind barb, and north arrow has been added per the map-making. Please refer to the author's track change file

Point 3.  The letters ( a-f) and numbers( 10)should not hide the pieces of information on the map. So, make it transparent for the background map information.  

Response 3. The letters has been made transparant. Please refer to the author's track change file

Point 4.  What is the meaning of  #10 and   other symbols? Please, avoid such confusion for your potential readers.

Response 4. The #10 does not have meaning, we had removed it.  Please refer to the author's track change file

Point 6. In Figure 10-13, the north arrow  should be placed on the maps per the cartographic/map-making/ principles, not outside the map. 

 Response 6. The north arrow has been placed on figure 10-13 per the map-making. Please refer to the author's track change file.
